# Impact of Siponimod on Enteric and Central Nervous System Pathology in Late-Stage Experimental Autoimmune Encephalomyelitis

**DOI:** 10.3390/ijms232214209

**Published:** 2022-11-17

**Authors:** Alicia Weier, Michael Enders, Philipp Kirchner, Arif Ekici, Marc Bigaud, Christopher Kapitza, Jürgen Wörl, Stefanie Kuerten

**Affiliations:** 1Institute of Neuroanatomy, Medical Faculty, University of Bonn, 53115 Bonn, Germany; 2Institute of Pathology, University of Bern, CH-3008 Bern, Switzerland; 3Institute of Human Genetics, University Clinic Erlangen, 91054 Erlangen, Germany; 4Novartis Institutes for BioMedical Research, CH-4002 Basel, Switzerland; 5Institute of Anatomy and Cell Biology, Friedrich-Alexander-Universität Erlangen-Nürnberg, 91054 Erlangen, Germany

**Keywords:** central nervous system, enteric nervous system, experimental autoimmune encephalomyelitis, fingolimod, multiple sclerosis, siponimod

## Abstract

Multiple sclerosis (MS) is an autoimmune disease of the central nervous system (CNS). Although immune modulation and suppression are effective during relapsing-remitting MS, secondary progressive MS (SPMS) requires neuroregenerative therapeutic options that act on the CNS. The sphingosine-1-phosphate receptor modulator siponimod is the only approved drug for SPMS. In the pivotal trial, siponimod reduced disease progression and brain atrophy compared with placebo. The enteric nervous system (ENS) was recently identified as an additional autoimmune target in MS. We investigated the effects of siponimod on the ENS and CNS in the experimental autoimmune encephalomyelitis model of MS. Mice with late-stage disease were treated with siponimod, fingolimod, or sham. The clinical disease was monitored daily, and treatment success was verified using mass spectrometry and flow cytometry, which revealed peripheral lymphopenia in siponimod- and fingolimod-treated mice. We evaluated the mRNA expression, ultrastructure, and histopathology of the ENS and CNS. Single-cell RNA sequencing revealed an upregulation of proinflammatory genes in spinal cord astrocytes and ependymal cells in siponimod-treated mice. However, differences in CNS and ENS histopathology and ultrastructural pathology between the treatment groups were absent. Thus, our data suggest that siponimod and fingolimod act on the peripheral immune system and do not have pronounced direct neuroprotective effects.

## 1. Introduction

Multiple sclerosis (MS) is a chronic autoimmune disorder of the central nervous system (CNS). The histopathology of MS is characterized by inflammation, demyelination, and axonal loss. Depending on the lesion topology, patients with MS can develop a wide range of symptoms [1]. Globally, 2.8 million people have MS, and disease incidence continues to increase. Women are affected twice as much as men, with a mean age at diagnosis of 32 years, which makes MS the most common nontraumatic cause of disability in young adults [2]. The etiology of MS is unclear; however, in addition to a genetic predisposition, e.g., the presence of certain *HLA-DRB1* alleles, vitamin D deficiency and smoking are also believed to be risk factors [3]. A longitudinal study has recently confirmed the causative relationship between Epstein–Barr virus infection and an increased susceptibility to MS [4]. Although MS is incurable, several options for disease-modifying treatment exist. Most interact with the immune system and are effective in patients with relapsing-remitting MS (RRMS) [5,6]. However, approximately 80% of patients with MS develop secondary progressive MS (SPMS) after a median time of 20 years after diagnosis [5,7,8], for which siponimod is the only approved drug [9,10]. Similar to its predecessor fingolimod, siponimod acts as a sphingosine-1-phosphate receptor (S1PR) modulator, inhibiting the egress of lymphocytes from secondary lymphoid organs. Treatment with fingolimod or siponimod leads to the internalization of S1PR1 in lymphocytes, which are then unable to react to the S1P gradient, trapping them inside the secondary lymphoid organs [11,12]. The resulting lymphopenia is associated with beneficial effects in the inflammation-driven RRMS stage. Siponimod does not only bind to S1PR1, but also to S1PR5, which is expressed by cells of the CNS, such as the oligodendrocytes and astrocytes [13,14]. In a *Xenopus laevis* toxin-induced demyelination model, siponimod was shown to improve remyelination through S1PR5 signaling [15]. In addition, the anti-inflammatory effects of the drug on astrocytes, microglia, and macrophages were associated with an effect on S1PR1 [16,17,18,19].

Interestingly, approximately 65% of patients with MS have gastrointestinal (GI) dysfunction. The most commonly described symptoms are constipation, dysphagia, dyspepsia, and fecal incontinence [20]. One-third of these patients have GI symptoms even before MS onset [21]. In 1983, the Expanded Disability Status Scale (EDSS) was established to monitor and quantify MS-related symptoms, which are clustered into seven functional systems. Since the introduction of the EDSS, the GI tract has been one of these functional systems, highlighting the awareness of GI symptoms in patients with MS for approximately 40 years [22]. In 2017, the enteric nervous system (ENS) was identified as an autoimmune target in experimental autoimmune encephalomyelitis (EAE), which is the most common animal model of MS [23]. The ENS is part of the autonomous nervous system (ANS) and the intrinsic nervous system of the GI tract. The ENS regulates functions such as mucus secretion, GI motility, and immunological responses. It shares many similarities with the CNS; all neurotransmitter classes of the CNS are also present in the ENS. Moreover, as the only part of the ANS, the ENS can initiate reflexes [24]. Human ENS contains 200–600 million neurons that are distributed into ganglia, which form two major plexi: the submucosal plexus is located inside the submucosal connective tissue underneath the epithelium and lamina propria (LP), whereas the myenteric plexus is located between the two smooth muscle layers of the tunica muscularis [25]. Using the MP4-induced EAE model, we demonstrated progressive, antibody-mediated myenteric plexus degeneration over the course of the disease [23]. MP4 is a fusion protein consisting of the human isoform of myelin basic protein (MBP) and the three hydrophilic domains of proteolipid protein. The immunopathology of MP4-induced EAE depends on both autoreactive T and B cells [26], and the disease course can be divided into an acute and chronic stage [27,28]. Consistently, Spear et al. showed reduced GI motility and fecal water content in myelin oligodendrocyte glycoprotein-, proteolipid protein-, and mouse spinal cord homogenate-immunized EAE mice [29]. Moreover, the preliminary data suggest that the ENS is affected in MS, because the loss of enteric nerve fibers and enterogliosis were detected in the myenteric plexus of gut resectates in patients with MS [23,30].

Here, we treated MP4-immunized mice with siponimod during the chronic stage of EAE, i.e., 50 d after peak disease—corresponding to day 70 after immunization—to examine the effects of the drug on the clinical disease severity and CNS and ENS histopathology. Because fingolimod was reported to be ineffective in primary progressive MS in the INFORMS trial [31], we evaluated the potential differences in the mode of action of siponimod and fingolimod in mice. This is the first study to investigate the effects of siponimod on an established disease and evaluate its effects on the ENS.

## 2. Results

### 2.1. Siponimod and Fingolimod Induce Peripheral Lymphopenia in MP4-Immunized Experimental Autoimmune Encephalomyelitis (EAE) Mice

Two individual cohorts of female C57BL/6J mice were immunized with MP4. EAE onset was 11.73 ± 0.3 days post immunization (d.p.i.), with the peak disease at approximately 18 d.p.i. (Table 1; Figure 1A,B). Treatment with siponimod or fingolimod started 50 d after peak EAE was reached and continued for 33 d. Within this treatment duration, no significant improvement of the clinical score was observed. To validate the treatment success, mass spectrometry analysis of the serum was conducted to determine the concentration of siponimod, fingolimod, or fingolimod phosphate (fingolimod-P; the active metabolite of fingolimod; Figure 1C). In siponimod-treated mice, a mean level of 1012.9 ± 53.4 nM was detected in the serum. In fingolimod-treated mice, only 57.6 ± 8.4 nM was detected in the serum, which could be explained by the phosphorylation of fingolimod. Consistently, a mean level of 618.9 ± 82.4 nM fingolimod-P was detected in the serum of fingolimod-treated mice. To test for drug efficacy, flow cytometry analysis of peripheral blood B and T cells was performed (Figure 1D,E). While vehicle-treated mice showed 7.08% ± 1.58% live B and 3.98% ± 1.10% live T cells, the mean percentage of B cells was reduced to 1.12% ± 0.32% in siponimod- and 1.21% ± 0.27% in fingolimod-treated mice. The mean percentage of T cells was reduced to 0.20% ± 0.06% in siponimod- and 0.33% ± 0.11% in fingolimod-treated mice. In summary, both drugs induced peripheral lymphopenia and were present in the serum in the expected concentrations [32].

### 2.2. The Enteric Nervous System Expresses Sphingosine-1-Phosphate (S1P) Receptors

MP4-induced EAE has been shown to manifest not only in the CNS, but also in the ENS, and ENS damage occurred even before the onset of the clinical signs of EAE [23]. To evaluate the potentially protective effect of siponimod on the ENS, a primary culture of enteric neurons and glial cells was established. The protocol was based on Smith et al. [33] and optimized for use with the Miltenyi gentleMACS^TM^ Octo Dissociator. To this end, the longitudinal muscular layer, with the attached myenteric plexus (LMMP), was removed from the complete intestine (from duodenum to rectum) and digested enzymatically and mechanically (Figure 2A). After 10 d in culture, a stable network of interacting cells was visible using light microscopy (Figure 2B). Immunofluorescence staining revealed the expression of typical enteric glial and neuronal markers—glial fibrillary acidic protein (GFAP) and βIII-tubulin, respectively. Interestingly, clearly separated neuron and glial cell staining, positive for only one of the markers, were only observed immediately after preparation until day 3 in culture. From day 5 onward, cells coexpressing GFAP and βIII-tubulin were present. Coexpression decreased from day 10 onward, increasing the proportion of βIII-tubulin single positive cells (Figure 2C). This observation is in line with Verìssimo et al., who reported that ex vivo isolated enteric glial cells could differentiate into neurons in culture [34].

Next, we evaluated the expression of *S1pr1*, *S1pr3*, *S1pr4*, and *S1pr5* using reverse transcribed cDNA from pure enteric neuronal or glial cell lines. Analysis using qualitative reverse transcription polymerase chain reaction (RT-PCR) revealed the expression of *S1pr1*, *S1pr4*, and *S1pr5* by enteric neurons (Figure 3A). In contrast, enteric glial cells expressed *S1pr3* (Figure 3B). These results were confirmed by quantitative real-time polymerase chain reaction (qRT-PCR; Figure 3C). Similar results were obtained at the protein level using the primary ENS cell culture system. Antibody staining revealed that S1PR1 and S1PR5 colocalized with βIII-tubulin. Antibody staining revealed that S1PR3 colocalized with GFAP (Figure 3D).

### 2.3. Siponimod and Fingolimod Do Not Have an Impact on the Pathology of the Enteric Nervous System (ENS) in Chronic Experimental Autoimmune Encephalomyelitis (EAE)

To study ENS pathology in MP4-immunized mice, immunohistochemical (IHC) staining was performed. Compared to nonimmunized mice, massive T cell infiltration into the jejunal LP was observed in the chronic MP4-EAE jejunum (Figure 4A). The LP of nonimmunized mice only contained 44.2 ± 28.8 T cells/mm^2^ in homeostasis; however, in EAE, the density increased 10-fold, resulting in 449.4 ± 45.9 (*p* = 0.034) cells/mm^2^ in vehicle-, 463.0 ± 37.0 (*p* = 0.012) cells/mm^2^ in siponimod-, and 370.1 ± 47.7 cells/mm^2^ (*p* = 0.11) in fingolimod-treated mice. In contrast, there was no difference in local B cell numbers (compare 71.9 ± 17.74 cells/mm^2^ in nonimmunized mice with 73.9 ± 9.97 in vehicle-, 82.9 ± 9.38 in siponimod-, and 72.57 ± 8.89 in fingolimod-treated mice; Figure 4A). Regarding the number of ionized calcium-binding adapter molecule 1 (IBA1)^+^ macrophages in the LP, there was a significant increase in EAE mice (compare 1.90% ± 0.34% in nonimmunized mice with 15.57% ± 1.68% (*p* < 0.001) in vehicle-, 12.38% ± 1.53% (*p* < 0.001) in siponimod-, and 13.90% ± 0.75% (*p* < 0.001) in fingolimod-treated mice; Figure 4B). Increased expression of GFAP (a plexus-specific enteric glial marker) was observed in the myenteric plexus in MP4-immunized mice (Figure 4C). Although 1.0% ± 0.30% of the muscularis area was stained positive for GFAP in nonimmunized mice, this value increased to 2.6% ± 0.31% (*p* < 0.001) in vehicle-, 2.9% ± 0.14% (*p* < 0.001) in siponimod-, and 2.2% ± 0.16% (*p* = 0.01) in fingolimod-treated mice. βIII-tubulin staining did not reveal statistically significant differences between nonimmunized and immunized mice (1.42% ± 0.35% muscularis area stained positive in nonimmunized mice versus 2.73% ± 0.68% in vehicle-, 3.90% ± 0.67% in siponimod-, and 3.32% ± 1.01% in fingolimod-treated mice; Figure 4D).

Staining of the colonic ENS revealed few lymphocytes residing in the LP. Therefore, no further analysis of CD3/B220 staining was performed. Differences between nonimmunized and immunized mice were evident for IBA1 and GFAP in a similar pattern (Figure 4E,F). Although 4.60% ± 1.53% of the IBA1^+^ area was present in nonimmunized mice, this value increased to 10.74% ± 0.36% (*p* = 0.0091) in vehicle-, 12.75% ± 1.24% (*p* = 0.0004) in siponimod-, and 12.88% ± 0.97% (*p* = 0.0003) in fingolimod-treated mice. GFAP values increased from 1.67% ± 0.32% for muscularis area in nonimmunized mice to 2.85% ± 0.21% (*p* = 0.04) in vehicle-, 2.94% ± 0.29% (*p* = 0.02) in siponimod-, and 3.26% ± 0.26% (*p* = 0.002) in fingolimod-treated mice. Finally, staining for βIII-tubulin revealed a significant decrease in the positively stained area in MP4-immunized mice; 6.78% ± 1.02% of muscularis area stained positive in nonimmunized mice compared with 2.30% ± 0.08% (*p* < 0.001) in vehicle-, 1.94% ± 0.19% (*p* < 0.001) in siponimod-, and 2.67% ± 0.25% (*p* < 0.001) in fingolimod-treated mice (Figure 4G). Taken together, enterogliosis was observed in both the jejunum and colon, accompanied by pronounced T cell infiltration into the jejunal LP and neuronal loss in the colonic myenteric plexus in chronic EAE. However, no significant differences in CD3, B220, IBA1, GFAP, and βIII-tubulin staining were present between the treatment groups.

Analysis using light microscopy revealed significant nerve fiber loss in the colonic myenteric plexus (Figure 4G); therefore, we performed additional ultrastructural analysis. Transmission electron microscopy analysis revealed pronounced pathology in all EAE mice (Figure 5A). Notable differences between nonimmunized and chronic EAE mice included axonal swelling and the presence of edematous gaps. ENS pathology mainly manifested as reduced axonal density, with a value of 7.38 ± 0.38 axons/µm^2^ in nonimmunized mice that decreased to 5.33 ± 0.40 (*p* < 0.001) in vehicle-, 5.45 ± 0.56 (*p* < 0.001) in siponimod-, and 4.80 ± 0.31 (*p* < 0.001) in fingolimod-treated mice (Figure 5B). Although a trend toward more axolytic axons in MP4-immunized mice was evident, statistical significance was not reached (0.81% ± 0.24% in nonimmunized mice, 1.75% ± 0.56% (*p* = 0.34) in EAE mice treated with vehicle, 1.53% ± 0.42% (*p* = 0.60) in EAE mice treated with siponimod, and 2.05% ± 0.54% (*p* = 0.08) in EAE mice treated with fingolimod) (Figure 5C). Overall, there was no statistically significant difference when comparing the three treatment groups.

### 2.4. Siponimod Has Limited Effects on Gene Expression in Enteric Neurons and Glial Cells

We performed single-cell RNA sequencing (scRNA-seq) of LMMP single cell suspensions from animals treated with vehicle, siponimod, or fingolimod. A total of 10 cell clusters could be distinguished in the integrated data set at a clustering resolution of 0.2. The clusters contained cells from all three conditions at comparable proportions (Figure 6A). Using a set of marker genes (top one is shown in Figure 6B) expressed by most cells of a certain cluster, and not expressed by most cells in the other clusters, different clusters could be identified (Figure 6C). Cluster 1 contained enteric neurons and glial cells. Cluster 2 contained fibroblasts, clusters 3 and 8 contained different subsets of T cells and natural killer cells. Cluster 4 contained colonic epithelial cells. Clusters 5 and 6 contained professional antigen presenting cells (APCs), while clusters 7, 9, and 10 contained endothelial cells. Because the aim of the study was to determine the potential neuroprotective role of siponimod, we focused our analysis on cluster 1. To this end, the top differentially expressed genes were analyzed between treatment groups. We defined a threshold of >|1.5| as differentially expressed, indicating that all genes with a log_2_ fold change (logFC) of >1.5 or <−1.5 were genes of interest, when there was also statistical significance with an adjusted *p*-value of < 0.05. In cluster 1 (Figure 6D), only one gene passed this threshold: *Dcn*, transcribing for the proteoglycan decorin, which is expressed by the extracellular matrix of many tissues and plays important roles in the protection against cancer [35]. *Dcn* was downregulated in the siponimod- compared with the fingolimod-treated group, with a logFC of −1.78. Compared to vehicle, both groups showed similar changes in *Dcn* expression, but in opposite directions, which was below the threshold. Siponimod-treated mice showed a downregulation of *Dcn* expression (logFC = −0.78), while fingolimod-treated mice showed an upregulation of *Dcn* expression (logFC = 1.00).

### 2.5. Siponimod and Fingolimod Do Not Affect Central Nervous System Pathology in Chronic Experimental Autoimmune Encephalomyelitis (EAE)

Next to the ENS, we examined CNS pathology in nonimmunized compared with MP4-immunized and treated mice. We focused on the spinal cord because of the prominent pathology observed in this region in MP4-induced EAE [36]. First, the pathology was examined with light microscopy. To this end, IHC staining for neuronal, glial, and immunological markers was performed. EAE mice showed a significantly higher number of infiltrating T cells in the spinal cord compared with nonimmunized mice. Only 0.25 ± 0.25 cells/mm^2^ were present in nonimmunized mice. This number increased to 14.2 ± 4.48 (*p* = 0.018) in vehicle-, 10.8 ± 4.1 (*p* = 0.053) in siponimod-, and 10.4 ± 2.48 T (*p* = 0.045) in fingolimod-treated mice (Figure 7A). Consistent with the results obtained from the intestine, B cell infiltration was absent; however, the number of IBA1^+^ cells increased in MP4-immunized mice. Compared to 0.29% ± 0.05% of the IBA1^+^ spinal cord area in nonimmunized mice, this number increased to 1.81% ± 0.33% (*p* = 0.09) in vehicle-, 1.88% ± 0.25% (*p* = 0.046) in siponimod-, and 2.15% ± 0.34% (*p* = 0.014) in fingolimod-treated EAE mice (Figure 7B). Anti-MBP antibody (clone SMI-99) staining for demyelination revealed fulminant pathology of the anterolateral white matter (Figure 7C). Quantification revealed a decrease by one-third of the MBP^+^ area in EAE mice (19.24% ± 1.76% (*p* = 0.30) positively-stained spinal cord area in vehicle-, 15.35% ± 1.43% (*p* = 0.004) in siponimod-, and 18.10% ± 1.21% (*p* = 0.046) in fingolimod-treated mice) compared with nonimmunized controls (29.91% ± 1.48%). Moreover, immunization induced reactive astrogliosis, shown by an increase in the GFAP^+^ area. The GFAP^+^ spinal cord area was 12.74% ± 1.07% (*p* = 0.037) in vehicle-, 12.54% ± 0.98% (*p* = 0.036) in siponimod-, and 9.98% ± 0.70% (nonsignificant) in fingolimod-treated mice compared with nonimmunized mice, which had 8.20% ± 1.00% GFAP^+^ area (Figure 7D). Furthermore, no differences between the treatment groups were observed in the number of Olig2/adenomatous-polyposis-coli (APC) double-positive cells (Figure 7E). Olig2 is an oligodendrocyte-specific transcription factor expressed at all stages of oligodendrocyte development, and APC is a tumor suppressor that is transiently expressed during myelination and remyelination [37]. Nonimmunized mice had 609.6 ± 28.7 Olig2/APC double-positive cells/mm^2^. Vehicle-, siponimod-, and fingolimod-treated EAE mice displayed 596.5 ± 14.7, 527.0 ± 23.8, and 544.8 ± 37.4 cells/mm^2^, respectively. βIII-tubulin staining revealed severe neuronal loss in MP4-immunized mice. In nonimmunized mice, 57.03% ± 6.84% of the spinal cord area stained positive for βIII-tubulin; however, in vehicle-, siponimod-, and fingolimod-treated mice, these values were 4.60% ± 1.26% (non-significant), 2.45% ± 0.61% (*p* = 0.010), and 3.59% ± 1.38% (*p* = 0.017), respectively. SMI-32 is a monoclonal antibody against the nonphosphorylated neurofilament heavy chain, which is expressed by pyramidal neurons, but is also detected in damaged axons [38,39]. Chronic MP4-immunized mice showed a significant increase in white matter SMI-32 staining (0.27% ± 0.10% (non-significant) in vehicle-, 0.37% ± 0.12% (*p* = 0.026) in siponimod-, and 0.40% ± 0.10% (*p* = 0.01) in fingolimod-treated mice) compared with nonimmunized mice (0.0005% ± 0.0004%) (Figure 7G). Taken together, IHC analysis revealed a pronounced pathology with light microscopy in the lumbar spinal cord of EAE mice. However, no statistically significant differences were observed between the treatment groups.

Moreover, we examined the spinal cord at the ultrastructural level. The numbers of normal axons, axons undergoing axolysis (Figure 8A, red arrows), and axons with a pathology or without the myelin sheath were quantified (Figure 8A, yellow arrows). In addition, we determined the value of the g-ratio, which is defined as the ratio between the area of the axon divided by the area of the nerve fiber (axon + myelin sheath). The value of the mean g-ratio in nonimmunized mice was calculated. A range defined as mean ± three standard deviations was considered “normal”. Nerve fibers with g-ratio values below or above the threshold were classified as demyelinating or remyelinating, respectively. EAE mice showed a significantly lower g-ratio than nonimmunized mice (Figure 8B, compare 0.52 ± 0.011 in nonimmunized mice to 0.46 ± 0.011 (*p* = 0.11) in vehicle-, 0.44 ± 0.008 (*p* = 0.004) in siponimod-, and 0.43 ± 0.018 (*p* = 0.014) in fingolimod-treated mice). No difference between treatment groups was observed indicating no effect of siponimod or fingolimod beyond spontaneous remyelination (Figure 8C). EAE mice displayed significantly more axons with a pathology or without the myelin sheath. Although nonimmunized mice had 2.04% ± 0.44% of such axons, this value increased to 10.39% ± 2.17% (*p* < 0.001) in vehicle-, 10.06% ± 1.28% (*p* < 0.001) in siponimod-, and 12.11% ± 2.12% (*p* < 0.001) in fingolimod-treated mice (Figure 8D). Moreover, less axons/µm^2^ were observed in some groups. Nonimmunized mice displayed a mean number of 0.23 ± 0.02 axons/µm^2^ in the myenteric plexus. In vehicle-, siponimod-, and fingolimod-treated mice, these numbers were 0.21 ± 0.02, 0.19 ± 0.01 (*p* = 0.004), and 0.17 ± 0.01 (*p* < 0.0001) axons/µm^2^, respectively (Figure 8E). The percentage of axolytic axons increased from 0.64% ± 0.20% in nonimmunized mice to 10.19% ± 1.68% (*p* < 0.001) in vehicle-, 11.79% ± 2.63% (*p* < 0.001) in siponimod-, and 12.61% ± 2.58% (*p* < 0.001) in fingolimod-treated mice (Figure 8F).

### 2.6. Siponimod and Fingolimod Do Not Induce Pro-Myelinating Gene Expression but Affect Immune Regulation

We next performed scRNA-seq from single cell suspensions of mechanically and enzymatically digested spinal cord. With a clustering resolution of 0.1, we could differentiate 11 cell clusters in the integrated data of all three treatments. The experimental conditions were evenly distributed between the clusters (Figure 9A). Using the top marker genes (top one is shown in Figure 9B), we could identify 9 clusters (Figure 9C). Clusters 1 and 2 contained microglial subsets. Clusters 3 and 10 contained endothelial cells. T cells were present in cluster 4, whereas collagenous connective tissue constituted cluster 5. Clusters 6 and 11 could not be identified and contained only low numbers of cells. Cluster 7 contained myelinating cells, including oligodendrocytes and Schwann cells. Astrocytes and ependymal cells were present in cluster 9, whereas cluster 10 contained pericytes. We focused on glial cell clusters 2, 7, and 8 to evaluate the neuroprotective effects of siponimod. In cluster 2, no gene passed the threshold value of logFC >|1.5| (Figure 9D). In cluster 7, *S100a9* was strongly upregulated in the fingolimod- compared with the vehicle-treated group (logFC = 2.02; Figure 9E). However, there was no change in expression in the siponimod-treated group compared with the vehicle-treated group (logFC = 0.19). *S100a9* encodes S100A9 or calprotectin, which is a proinflammatory calcium- and zinc-binding protein that is upregulated in the serum of patients with MS [40]. In cluster 8, 14 differentially expressed genes passed the threshold (Figure 9F). The logFC values are shown in Table 2.

We found that *S100a9* was upregulated; however, this time upregulation was significantly higher in siponimod- than fingolimod-treated mice. A similar expression pattern was observed for *S100a8*, which forms a dimer with the gene product of *S100a9* [40]. *Camp,* which encodes cathelicidin antimicrobial peptide (CAMP) [41], was significantly upregulated in siponimod-treated mice compared with vehicle- and fingolimod-treated mice. A similar upregulation was observed for neutrophilic granule peptide (*Ngp*), lysozyme C2 (*Lyz2*), lipocalin-2 (*Lcn2*), chitinase-like protein 3 (*Chil3*), and serum amyloid A-3 (*Saa3*). The three complement C1q subunits *C1qc*, *C1qb*, and *C1qa*, as well as *Cd74*, showed a trend toward upregulation when compared with vehicle-treated mice, but passed the threshold only when comparing siponimod- and fingolimod-treated mice. Oligodendrocyte transcription factor 1 (*Olig1*) was the only gene that was downregulated when siponimod- and fingolimod-treated mice were compared.

## 3. Discussion

In this study, mice were immunized with MP4 to induce EAE and fed with either siponimod-, fingolimod-, or vehicle-loaded food for 33 d. Subsequent analysis of ENS and CNS pathology revealed no statistically significant differences between the treatment groups.

Siponimod and fingolimod are S1PR modulators. Siponimod only binds to S1PR1 and S1PR5, while fingolimod additionally binds to S1PR3 and S1PR4. The higher specificity of siponimod is linked to less severe adverse effects because, for example, the occurrence of cardiovascular side effects of fingolimod is mediated by S1PR3 [42]. Both drugs mainly induce peripheral lymphopenia by binding to S1PR1, which renders lymphocytes unable to respond to the S1P gradient [10,11]. Moreover, siponimod and fingolimod have anti-inflammatory effects on CNS-resident cells. Siponimod pretreatment reduced the secretion of the proinflammatory cytokines IL-6 and CCL-5 in the tumor necrosis factor-treated murine microglial cell line BV-2 [19]. Consistently, siponimod treatment reduced IL-6 secretion in lipopolysaccharide-induced mouse microglial cells and astrocytes [17]. However, human astrocytes, which did not react to lipopolysaccharide-induced stimulation, increased IL-6 secretion after treatment with tumor necrosis factor alpha and IL-17. Here, pretreatment with siponimod had no effects. According to the same study, the siponimod-induced phosphorylation of the serine/threonine-kinases ERK and AKT in astrocytes as well as increased Ca^2+^ signaling was associated with pro-survival pathways in mice and humans [17]. Siponimod pretreatment of induced pluripotent stem cell-derived astrocytes that were exposed to IL-1, IL-17, or S1P reduced neurodegeneration by inhibiting the translocation of the transcription factor NF-κB-p65, which amplifies inflammation and neurodegeneration. A similar effect could be achieved by treatment with fingolimod [16]. Siponimod has been shown to have beneficial effects on the blood–brain barrier. In an in vitro model of the blood–brain barrier, siponimod diminished the migration rate of peripheral blood mononuclear cells by reducing the expression of CCL2 in the astrocytes [43]. Consistently, the blocking of lymphocyte egress by siponimod and fingolimod was shown to be mediated by the CCL2 pathway [18].

Considering the treatment of progressive MS, neuroprotective and remyelinating effects of siponimod and fingolimod are desirable. Toxin-induced demyelination models have the advantage that they lack an autoimmune response [44]. In organotypic slice cultures, both drugs could attenuate demyelination induced by psychosine in vitro [17,45]. In vivo, siponimod was shown to improve remyelination in the *X. laevis* tadpole, in which demyelination had been induced by metronidazole. This effect was achieved through S1PR5 because S1PR5-knockout animals did not exhibit remyelination after siponimod exposure. Although fingolimod was not as potent as siponimod, remyelination was significantly improved compared with the controls in the same study [15]. In the cuprizone-induced mouse demyelination model, siponimod-treated mice did not lose weight compared with control mice. Longitudinal analysis of the corpus callosum, through magnetic resonance imaging, revealed a slight, but significant increase in remyelination one week after cuprizone washout, which was however, not significant one week later [46].

Mirroring both the inflammatory and demyelinating aspect of MS, EAE is the most widely used animal model for MS research. Siponimod has been tested in various EAE models. EAE severity could be reduced using osmotic minipumps to deliver siponimod directly into the cerebrospinal fluid. However, in one study, the minipumps were installed before EAE induction and even the lowest concentration reduced peripheral T cells by 20%, suggesting that the protective effects of siponimod on the CNS were due to effects on the peripheral immune system [19]. Minipumps were also used in a model of focal EAE, where gray matter lesions were induced by injecting proinflammatory cytokines into the brain. Minipumps were installed in the same operation as focal gray matter lesions were induced 10 d after EAE induction. In this study, siponimod was not able to reduce the clinical score and only slightly decreased lymphocyte infiltration into the CNS [47]. In a spontaneous EAE model, preventive treatment with siponimod inhibited the formation of ectopic lymphoid tissue and decreased the degree of demyelination. However, when mice were treated after having reached an EAE score of three, only minor effects on spinal cord demyelination were observed, whereas pronounced effects on meningeal ectopic lymphoid tissue formation were evident [48], again highlighting a predominant effect of siponimod on the immune response. In another model, subpial gray matter lesions and the formation of ectopic lymphoid structures were induced by transferring T_H_17 cells of mice immunized with proteolipid protein peptide 139–151 into naïve mice. In untreated mice, gray matter lesions and ectopic lymphoid structures started forming on day 5 after transfer. When mice were treated with siponimod from day 3 after transfer, EAE could be completely suppressed. Treatment from day 5 onward reduced the symptoms. Treatment from day 8 onward was not associated with clinical improvement [49]. A recent study was the first to compare the early- and late-stage effects of siponimod on optic nerve demyelination in myelin oligodendrocyte glycoprotein peptide 35–55-induced EAE. Although early treatment was successful, no positive effect of siponimod was observed when treatment started 30 d after immunization [46].

In line with Dietrich et al. [46], our study did not show any differences in clinical EAE parameters and histopathology comparing all three treatment groups. MP4-induced EAE is characterized by demyelination and axonal damage, with increasing severity over the course of the disease [27]. At the same time, there is ongoing inflammation in this model, which becomes compartmentalized as B cell aggregates within the CNS in late-stage disease [50]. Over the course of the disease, neurodegeneration becomes a prominent aspect of EAE immunopathology. The clinically evident EAE score is typically caused by lumbar spinal cord damage, which occurs immediately after disease onset and is in part irreversible due to axonal pathology [28]. We have previously observed remyelination to occur only in late-stage disease in the C57BL/6 model [28]. Hence, we hypothesized that any remyelinating effect of a drug should be best observable in late-stage disease. In addition, several studies have already addressed the effects of siponimod and fingolimod in the early disease [19,46,48,49,51], so that our study was designed to close a gap by adding results for the chronic disease stage. Hence, we chose to initiate treatment 70 d after immunization to focus on primary neuroregenerative effects of siponimod and fingolimod.

We evaluated the ENS and spinal cord using both light microscopy and ultrastructural analysis. Although oligodendrocytes express S1PR5, which is believed to be the receptor that mediates the beneficial effects of siponimod [15], our data demonstrate that S1PR5 is not expressed by enteric glial cells. This might explain the lack of response of enteric glial cells to siponimod and fingolimod at the mRNA level. ScRNA-seq only revealed *Dcn* to be differentially expressed in the cell cluster containing enteric neurons and glial cell. *Dcn* encodes the proteoglycan decorin and was downregulated in siponimod- and upregulated in fingolimod-treated mice. However, the change in expression did not reach the threshold of >|1.5| logFC compared with that in vehicle-treated mice, but only between the two treated groups. Decorin upregulation together with upregulation of other extracellular matrix (ECM) proteins has previously been observed in the Theiler’s murine encephalomyelitis virus model of MS. Here, deposition of ECM proteins paralleled the development of astrogliosis, hence suggesting an astrocytic origin of these ECM proteins. The accumulation of ECM proteins was believed to be one reason for remyelination failure in this model [52]. In a rat model of spinal cord injury, oral treatment with the drug C286, a retinoic acid receptor-beta agonist, induced a neural expression of decorin, which led to an increased differentiation of NG2^+^ oligodendrocyte precursor cells promoting myelination [53]. Our scRNA-seq results show changes in decorin expression in the cluster containing both enteric glial cells and neurons. Future studies will have to determine the exact source of the decorin expression in the ENS and whether an up- or downregulation has any effects on EAE outcome.

We performed IHC staining for myelin and oligodendrocytes in the spinal cord. The MBP-positive area was significantly decreased in late-stage EAE compared with that in nonimmunized mice, but no differences were present between the treatment groups. Additionally, APC/Olig2 double-positive cells, which were reported to transiently increase in number after demyelination [37], were not more abundant in EAE mice than in nonimmunized mice. ScRNA-seq revealed that only one gene was differentially expressed in the oligodendrocyte and Schwann cell cluster of the spinal cord: S100A9 is a calcium-binding protein that often forms a heterodimer with S100A8 [40]. S100A9 was significantly upregulated in fingolimod-, compared with vehicle- and siponimod-treated, mice. S100A9 levels have been found to increase in the serum of patients with MS and the protein may also be involved in the pathology of other CNS diseases, such as Alzheimer’s disease and stroke. Interestingly, S100A9 has been shown to activate microglia that contribute to oligodendrocyte damage [40], which contradicts a neuroprotective effect of fingolimod. Several genes were differentially regulated in the cell cluster containing astrocytes and ependymal cells in siponimod-treated mice. Of these, *Chil3* was upregulated in the siponimod-treated group. Chil3 is associated with anti-inflammatory and promyelinating functions. Chil3 was reported to induce oligodendrogenesis, and the silencing of *Chil3* increased the severity of EAE [54]. Other genes that were significantly upregulated in the siponimod-treated group are considered proinflammatory, i.e., *S100a9, S100a8*, and *Camp*. CAMP production by astrocytes was reported to promote neuroinflammation through crosstalk with microglia in EAE. In addition, EAE symptoms worsened after intrathecal injection of CAMP [41]. *Ngp* (expressed in neutrophils [55]) and *Lyz2* were upregulated by the astrocyte and ependymal cell cluster in siponimod-treated mice. Although NGP is thought to be proinflammatory and involved in cancer development [56], *Lyz2* is assigned a bacteriolytic function. In a mouse model of Niemann–Pick disease type C1, which is a genetic neurodegenerative disease, *Lyz2* was shown to be upregulated in different cell types of the brain, including astrocytes [57]. Further, *Lcn2* was upregulated in the siponimod-treated group. Lcn-2 was found to be upregulated in EAE, and increased Lcn-2 levels were measured in the serum and cerebrospinal fluid of patients with MS. Monocytes and reactive astrocytes were identified as the main source of Lcn-2 in the spinal cord [58]. In addition, Lcn-2 expression was reported to be higher in progressive MS compared with RRMS, and Lcn-2 levels in the cerebrospinal fluid correlated with neurofilament light chain levels, i.e., axonal damage. In the same study, in vitro treatment of myelinating cultures with Lcn-2 inhibited myelination in a dose-dependent manner [59]. Finally, we found that *Saa3* was upregulated. *Saa3* was shown to be upregulated in microglia and monocyte-derived macrophages in EAE, which advanced a feed-forward loop toward a T_H_17-mediated immune response driving EAE pathology [60].

Taken together, excluding *Chil3*, all genes that were upregulated in the siponimod-treated group were proinflammatory and associated with a worse MS outcome. Compared to the vehicle-treated group, only *Car3* was significantly upregulated in fingolimod-treated mice in the astrocyte and ependymal cell cluster. *Car3* encodes carbonic anhydrase 3, which catalyzes the interconversion between hydrogen carbonate and carbon dioxide + water. A study from 1989 reported the expression of carbonic anhydrases in fresh MS lesions [61], but further studies on the role in MS are lacking. However, *Car3* upregulation has been associated with anoxic stress in the cerebral cortex [62].

In conclusion, scRNS-seq data suggest a proinflammatory effect of siponimod on astrocytes and ependymal cells and no effect on oligodendrocytes and Schwann cells. Fingolimod strongly upregulated the proinflammatory gene *S100a9* in oligodendrocytes and Schwann cells. In the astrocyte and ependymal cell cluster, it upregulated *Car3*, whose connection to MS is unknown. It is unclear why the upregulation of proinflammatory genes in the siponimod group was not associated with a worse clinical EAE outcome and more severe histopathology. Reasons for this discrepancy might be that the upregulation of genes on the mRNA level did not translate into the protein level or that the extent of upregulation was not pronounced enough to induce any clinical or histopathological effect.

The proinflammatory effects of siponimod and fingolimod strongly contradict the clinical evidence obtained from patients with MS. In the phase II BOLD study, siponimod treatment in patients with RRMS induced a dose-dependent reduction of combined unique active lesions by 82% compared with placebo [63]. The phase III FREEDOMS trial showed a reduction in the annual relapse rate of up to 60% in fingolimod-treated patients with RRMS [64]. These studies demonstrate that both drugs are effective in RRMS. Regarding progressive MS, fingolimod has only been tested in PPMS. In the phase III INFORMS trial, fingolimod was shown to have no significant impact on disease progression compared with a placebo [31]. In contrast, siponimod showed significant effects on the disease progression in patients with SPMS in the phase III EXPAND study [9] by reducing the risk of confirmed disability progression after 6 months by 26%, compared with placebo. Notably, in the EXPAND trial siponimod was more effective in patients with residual inflammatory activity; therefore, the sponimod-induced effects might have been related to immune modulation rather than neuroprotection [9].

In conclusion, our findings did not reveal any significant beneficial effects of siponimod and fingolimod on ENS and CNS pathology using light microscopy, ultrastructural analysis, and analysis of mRNA levels. Siponimod even induced a proinflammatory mRNA expression profile in the spinal cord in late-stage EAE. Although our data were generated using the EAE model and extrapolation of data to MS should be done with caution, the results suggest that sponimod is not an ideal drug candidate for the induction of remyelination in progressive MS.

## 4. Materials and Methods

### 4.1. Experimental Autoimmune Encephalomyelitis (EAE) Induction, Treatment and Verification of Treatment Success

Female, 9–12-week-old C57BL/6J mice (Charles River, Sulzfeld, Germany; strain code 632) were kept under specific pathogen-free conditions at the Franz-Penzoldt-Zentrum of the University Hospital Erlangen. For EAE induction, 2 mg/mL MP4 (Alexion Pharmaceuticals, Boston, MA, USA) were mixed in a 1:1 ratio with complete Freund’s adjuvant, consisting of nine parts paraffin oil (Sigma-Aldrich, St. Louis, MO, USA (Cat. No. 18512)), one part mannide monooleate (Sigma-Aldrich (Cat. No. M8819)), and 5 mg/mL *Mycobacterium tuberculosis* H37Ra (BD Difco, Franklin Lakes, NJ, USA (Cat. No. 231141)). From this mixture, 100 µL were injected subcutaneously into both sides of the flanks, resulting in a total dose of 200 µg MP4 per mouse. On the day of immunization, and 24 h later, mice received an intraperitoneal injection of 120 ng pertussis toxin (Hooke Laboratories, Lawrence, KS, USA (Cat. No. BT-0105)) diluted in 100 µL sterile PBS (Thermo Fisher, Waltham, MA, USA (Cat. No. 10010)). Two independent cohorts of mice were immunized and used for different purposes. Cohort 1 (*n* = 20) was used for perfusion fixation and tissue embedding in paraffin or epoxy resin, respectively. Cohort 2 (*n* = 18) was used for the isolation of fresh tissue for scRNA-seq and PCR. All experiments were performed according to established protocols that were approved by the Government of Lower Franconia (“Regierung von Unterfranken”; license no. 55.2-2531.01-91/14) and complied with the German Law on the Protection of Animals, the “Principles of Laboratory Animal Care” (NIH publication no. 86-23, revised 1985), as well as the ARRIVE guidelines for reporting animal research [65]. Mice were scored daily according to the standard EAE scoring scale with: 0 = no symptoms, 1 = limp tail, 2 = hindlimb weakness, 3 = hindlimb paralysis, 4 = hindlimb paralysis + forelimb weakness or paralysis, and 5 = death. Fifty days after the peak score was reached, the mice were treated with siponimod, fingolimod, or vehicle. To this end, mice were fed with food loaded with 20 mg/kg siponimod (Novartis, Basel, Switzerland; *n* = 8 in cohort 1, *n* = 6 in cohort 2) or fingolimod (Novartis; *n* = 8 in cohort 1, *n* = 6 in cohort 2). A control group (*n* = 6 in each cohort) received sham food (=vehicle). Mice were fed ad libitum for 33 d subsequently. Different group sizes resulted from different EAE incidences.

To evaluate the treatment success, approximately 200 µL blood were drawn on the day before the end of the experiment. To avoid clotting, the blood was treated with sodium heparin (Ratiopharm, Ulm, Germany). For flow cytometry analysis, samples were incubated with 1 µL each of anti-CD3e-BV510 (Becton Dickinson, Franklin Lakes, NJ, USA (Cat. No. 563024)), anti-CD19-APC (BioLegend, San Diego, CA, USA (Cat. No. 115511)), and a fixable viability stain 450 (Becton Dickinson (Cat. No. 562247)) for 30 min at room temperature. Then, erythrocytes were lysed using 3 mL lysis buffer (BioLegend (Cat. No. 420401)) per sample. Blood samples were centrifuged at 500× *g* for 5 min. Pellets were resuspended and washed in 3 mL FACS Flow solution (Becton Dickinson (Cat. No. 342003)) and centrifuged. Pellets were resuspended in 150 µL FACS Flow solution and analyzed using the Beckman Coulter CytoFLEX cytometer. For analysis, singlets were gated using forward scatter (area) vs. forward scatter (height). Live cells were gated using forward scatter (area) vs. fixable viability stain 450. T and B cells were then gated using CD3e-BV510 bs. CD19-APC.

### 4.2. Perfusion Fixation and Tissue Embedding

For microscopy analysis, tissues were fixed by intracardial perfusion. Mice were euthanized using CO_2_, the abdominal and thoracic cavities were opened, the inferior vena cava was cut, and blood was collected for analysis of the serum. A venipuncture cannula was placed into the left ventricle and the mice were first shortly perfused with Ringer’s solution (B. Braun, Melsungen, Germany (Cat. No. 3570030). Then, perfusion fixation was performed using 4% paraformaldehyde (PFA; Carl Roth, Karlsruhe, Germany (0335)) in Sørensen’s phosphate buffer (SPB) for 20 min.

For the subsequent paraffin embedding, tissues were immersion-fixed in 4% PFA in SPB overnight at 4 °C. Tissues were then rinsed with SPB for 2 d, dehydrated in an ascending ethanol series and xylene, and embedded in Paraplast Plus^®^ (Carl Roth, Karlsruhe, Germany (X881)). Paraffin sections were cut at 5-µm thickness using the HistoCore MULTICUT microtome (Leica, Wetzlar, Germany).

For the subsequent epoxy embedding, tissues were immersion-fixed in a buffer consisting of 4% PFA, 2.14% sodium cacodylate (SERVA, Heidelberg, Germany (Cat. No. 15540)), 4% glutaraldehyde (Carl Roth (Cat. No. 4157.1)), and 0.002% picric acid (AppliChem, Darmstadt, Germany (Cat. No. A2520)). Tissues were then rinsed with a buffer containing sodium cacodylate. Specimens were stained using osmium tetroxide (emsdiasum, Hatfield, USA (Cat. No. 19190)) and potassium ferricyanide (Merck, Darmstadt, Germany (Cat. No. 4115149)). Tissues were dehydrated using an ascending ethanol series and acetone and embedded in Epon (Carl Roth). Ultrathin sections of 80 nm were cut and contrasted with uranyl acetate (emsdiasum (Cat. No. E22400)) and lead citrate (Polysciences Inc., Warrington, PA, USA (Cat. No. 25350-100)). Sections were analyzed using a Zeiss EM 906 transmission electron microscope (Carls Zeiss, Jena, Germany) at a cathode voltage of 60 kV.

### 4.3. Mass Spectrometry

Blood obtained during dissection for perfusion was kept at 4 °C until clotting, and then centrifuged at 4 °C and 10,000× *g* for 10 min. Serum was transferred into fresh tubes and sent to the Novartis Institutes for BioMedical Research for mass spectrometry analysis of siponimod, fingolimod, and fingolimod-P levels. Serum samples were diluted 1/1 with a 1/1 (*v*/*v*) mixture of acetonitrile and distilled water. Then, 250 ng/mL of stable isotype-labeled internal standard was added. Proteins were precipitated by adding a mixture of acetonitrile/methanol and trichloroethene (4/3/3 by volume). Samples were sonicated for 10 min and centrifuged at 16,100× *g* for 10 min. The upper layer was transferred to an Eppendorf LoBind tube (Eppendorf, Hamburg, Germany). The precipitation step was repeated and the transferred layers were combined. The samples were dried using the SpeedVac evaporator (Thermo Fisher) at 43 °C. Pellets were redissolved in 5 mM ammonium formate in methanol containing 0.2% formic acid and centrifuged at 16,100× *g* for 5 min. The supernatant was transferred to HPLC microvials. For analysis, an Agilent 1290 II UPHLC system couple, consisting of a binary high-pressure pump with an integrated degasser and static mixer, a multisampler, and a heated column compartment, coupled to the 6495 QqQ mass spectrometer (Agilent GmBH, Waldbronn, Germany) was used. The system was controlled, and data were processed using the MassHunter Workstation. For separation, 2 µL of each sample were injected into the ZORBAX RRHD Eclipse Plus C18, 2.1 × 50 mm column filled with 1.8-µm particles (Agilent, 959757-902), held at 40 °C. For separation, a two-step linear gradient from 20% to 50% B within 0.2 min and from 50% to 100% B within 1.7 min was used. Solvent A was 0.2% formic acid in water and solvent B 0.2% was formic acid in acetonitrile. The flow rate was kept constant at 500 µL/min.

For detection, column effluent was guided directly to the electrospray Jet Stream source of the triple quadrupole MS with parameters optimized for BAF312. The gas temperature was 210 °C at 16 L/min, the nebulizer pressure was 25 psi, and the sheath gas temperature was 350 °C at 12 L/min. Compound and internal standards were detected as their [MH]+ ions, with the MRM transition 517.3 *m*/*z* > 159.0 *m*/*z*, 416.2 *m*/*z* for BAF312. Data processing was based on compound/internal standard ratio of the extracted ion chromatograms.

For calculation, an external standard method using non-weighted, linear regression was used. The quality parameters of the method are summarized in the Table 3 below:

### 4.4. Primary ENS Culture

Female adult C57BL/6J mice were used for isolating the myenteric plexus cells. For each culture, one mouse was euthanized using CO_2_. The abdominal and thoracic cavities were opened, the inferior vena cava was cut, a venipuncture cannula was placed into the left ventricle, and the mouse was shortly perfused with Hanks’ balanced salt solution (HBSS) without Ca^2+^ and Mg^2+^ (HBSS^−^). The entire intestine, from distal to the pyloric sphincter to the proximal end of the anus, was dissected. The intestine was cut into four pieces and the cecum was removed. Fecal matter was removed and rinsed out using HBSS^−^. The tissues were then cut into 3–4-cm long pieces, and the LMMP was dissected by threading the tissue onto a glass rod, incising it longitudinally and rubbing off the LMMP with a sterile cotton swab wetted with HBSS^−^. The tissues were minced using scissors and transferred into a Miltenyi C Tube (Miltenyi Biotec 130-096-334) containing 5 mL HBSS with 5% fetal calf serum (FCS). Then, 13 mg type II collagenase and 3 µg bovine serum albumin (BSA) were added, and the tissues were digested with the gentleMACS^TM^ Octo Dissociator (Miltenyi Biotec), using the 37C_NTDK_1 program two times. Cells were centrifuged at 300× *g* for 8 min. The pellet was resuspended in 5 mL HBSS containing 0.05% trypsin and digested using 37C_NTDK_1 for 5 min. The trypsin reaction was stopped using prewarmed DMEM-F12 medium containing 10% FCS and 1% penicillin–streptomycin (=rinse medium). Cells were then passed through a 70-µm cell strainer. The strainer was rinsed with rinse medium and cells were centrifuged at 300× *g* for 5 min. Cells were resuspended in Neurobasal A medium (Thermo Fisher 10888022) containing 1% B-27 (Thermo Fisher (Cat. No. 17504001)), 1% FCS, 1% L-glutamine (Thermo Fisher (Cat. No. 25030081)), 1% penicillin–streptomycin, and 10 ng/mL glial-derived neurotrophic factor (Thermo Fisher (Cat. No. PHC7054)). Cells were seeded into 12 wells of a 24-well plate, which were coated with poly-D-lysine (Sigma-Aldrich (Cat. No. P6407)) and mouse laminin (Thermo Fisher (Cat. No. 23017-015)) in a volume of 800 µL/well. The cells were kept at 37 °C and 5% CO_2_. One half of the medium was changed every second day.

The cells were cultured for 10 d, washed one time with prewarmed PBS, and fixed with prewarmed 4% PFA in PBS for 10 min. Cells were washed with PBS and permeabilized with 0.1% Triton-X 100 (MP Biomedicals (Cat. No. 194854)) in TBS for 30 min at room temperature. Nonspecific antibody binding was blocked using 5% donkey serum (Biozol (Cat. No. END9010-10)) in TBS + 0.05% Tween-20 (Sigma-Aldrich (Cat. No. P2287), TBST). Cells were incubated overnight at 4 °C with primary antibodies diluted in 0.5% donkey serum in TBST. The following antibodies and dilutions were used: mouse anti-βIII-tubulin (Thermo Fisher (Cat. No. MA1-118); 1:200), chicken anti-GFAP (abcam (Cat. No. ab4674); 1:800), rabbit anti-S1PR1 (Thermo Fisher (Cat. No. MA5-32587); 1:100), rabbit anti-S1PR3 (Thermo Fisher (Cat. No. PA5-23225); 1:200), and rabbit anti-S1PR5 (Thermo Fisher (Cat. No. PA5-100934); 1:500). The cells were washed with TBST three times. The following secondary antibodies were used in 1:400 dilution: donkey anti-chicken Cy2 (Jackson ImmunoResearch, Cambridgeshire, UK (Cat. No. 703-225-155)), donkey anti-rabbit Cy3 (Jackson ImmunoResearch (Cat. No. 711-165-152)), donkey anti-rat Cy2 (Jackson ImmunoResearch (Cat. No. 712-225-150)), and goat anti-mouse Alexa Fluor 488 (abcam (Cat. No. ab150113)). Cells were incubated with secondary antibodies (diluted in 0.5% donkey serum in TBST) for 1 h at room temperature; washed with TBST, TBS, and distilled water; and embedded with Fluoroshield mounting medium with DAPI (abcam (Cat. No. ab104139)). Staining was analyzed using the Leica DMI8 Thunder imager equipped with a K5-14401790 camera (Leica) and the filter cube DFT51010 that consisted of DAPI (excitation filter [EX] 391/32, dichroic mirror [DC] 415, suppressor filter [SF] 435/30), Cy2 (EX 479/33, DC 500, SF 519/25), and Cy3 (EX 554/24, CD 572, SF 594/32).

### 4.5. PCR

For the characterization of *S1pr* RNA expression in ENS cells, RNA from pure cell lines of enteric neurons and enteric glial cells was used. The murine enteric neuronal RNA was a kind gift from Prof. Shanthi Srinivasan, from the Division of Digestive Diseases at Emory University, 30307 Atlanta, USA. The rat enteric glial RNA was a kind gift from Prof. Thilo Wedel, from the Anatomical Institute of the Christian–Albrechts-Universität zu Kiel, 24098 Kiel, Germany. The organs used as positive controls were immediately snap-frozen in liquid nitrogen after dissection. For RNA isolation, the tissues were ground up with a mortar and pestle in liquid nitrogen. The tissue powder was transferred into a nuclease-free tube containing TRIzol (Thermo Fisher (Cat. No. 15596026)) and incubated for 5 min at room temperature. Samples were shaken after the addition of chloroform (Sigma-Aldrich (Cat. No. 25668)) and centrifuged at 4 °C and 12,000× *g* for 15 min. The upper, RNA-containing phase was transferred into a fresh nuclease-free tube, incubated with 2-propanol (Sigma-Aldrich (Cat. No. I9516)) for 10 min at room temperature, and centrifuged at 4 °C and 12,000× *g* for 10 min. The pellet was washed with 75% ethanol (AppliChem (Cat. No. A1613)), centrifuged at 4 °C and 7500× *g* for 5 min, air dried, dissolved in nuclease-free water (Carl Roth (Cat. No. T143)), and incubated at 60 °C on a heat block for 10 min.

Reverse transcription to cDNA was performed using the High Capacity cDNA Reverse Transcription Kit (Thermo Fisher (Cat. No. 4368814)), according to the manufacturer’s protocol. For qualitative reverse transcription PCR (RT-PCR) the Red MasterMix from Genaxxon (Genaxxon, Ulm, Germany (Cat. No. M3029)) was used according to the manufacturer’s protocol with the following setup (Table 4) and primers (Table 5).

PCR products were transferred to a 1.5% agarose gel. Gel electrophoresis was performed at a 100 V constant for 60 min. For size comparison, a 100-bp gene ladder was used (New England BioLabs (Cat. No. N3231L)).

For quantitative real-time PCR (qRT-PCR), the same cDNA was used as that for RT-PCR. TaqMan Fast Advanced Master Mix (Thermo Fisher (Cat. No. 4444556)) was used according to the manufacturer’s protocol. For each reaction, 10 ng was used. The following tables show the cycling setup (Table 6) and assays (Table 7) used.

### 4.6. EM Image Analysis

Ten images of the colonic myenteric plexus per section were imaged at 10,000× magnification. Five of these images were randomly chosen for image analysis. The ventrolateral tract of the lumbar spinal cord was imaged at two magnifications. Twenty images at 3597× magnification were acquired, of which 5 were randomly chosen for quantification of axonal and myelin pathology. An additional 20 images were taken at 6000× magnification, of which 10 were randomly chosen for g-ratio analysis. Thirty axons per image were measured, resulting in a total of 300 axons per mouse.

### 4.7. IHC

Intestinal and spinal cord tissue sections were deparaffinized with xylene and rehydrated using a descending 2-propanol series. Heat-mediated antigen retrieval was performed using 0.1 M sodium citrate at pH 6.0 (Merck (Cat. No. 106448)) for GFAP, βIII-tubulin, CD3 + B220, SMI-32, and SMI-99 staining or 0.1 M Tris-EDTA at pH 9.0 (Carl Roth (Cat. No. 5429/CP87)) for IBA1 and Olig2 + APC staining. Nonspecific binding was blocked by incubating the sections with 5% milk powder (Heirler, Radolfzell, Germany (Cat. No. 3030)) for GFAP, IBA1, CD3 + B220, SMI-32, SMI-99, and Olig2 + APC or 5% donkey serum (Dianova, Hamburg, Germany (Cat. No. 017-000-121)) for βIII-tubulin in TBST at room temperature for 1 h. Sections were incubated with primary antibodies diluted in the corresponding blocking solution overnight at 4 °C. The following antibodies and dilutions were used: chicken anti-GFAP (abcam, Cambridge, UK (Cat. No. ab4674); 1:500), rabbit anti-IBA1 (FUJIFILM Wako, Neuss, Germany (Cat. No. 019-19741); 1:500), rabbit anti-βIII-tubulin (abcam (Cat. No. ab18207); 1:2000), rabbit anti-CD3 (abcam (Cat. No. ab16669); 1:500), rat anti-B220 (Thermo Fisher (Cat. No. 14-0452); 1:200), mouse anti-non-phosphorylated neurofilament light chain (clone SMI-32, Calbiochem (Merck), (Cat. No. NE-1023); 1:500), mouse anti-MBP (clone SMI-99, BioLegend (Cat. No. 808401); 1:800), rabbit anti-Olig2 (Abcam (Cat. No. ab136253); 1:200), and mouse anti-APC (Calbiochem (Cat. No. OP-80); 1:200). Sections were washed two times with TBST and one time with TBS, for 10 min each, and incubated with secondary antibodies diluted in the corresponding blocking buffer at room temperature for 1 h. The following secondary antibodies and dilutions were used: donkey anti-chicken Cy2 (Jackson ImmunoResearch, Cambridgeshire, UK (Cat. No. 703-225-155); 1:200), donkey anti-rabbit Cy3 (Jackson ImmunoResearch (Cat. No. 711-165-152); 1:200), donkey anti-rat Cy2 (Jackson ImmunoResearch (Cat. No. 712-225-150); 1:200), and goat anti-mouse Alexa Fluor 488 (abcam (Cat. No. ab150113); 1:200). The sections were washed, incubated for 3 min with Hoechst 33258 (Merck B2882) diluted 1:1000 in TBST, rinsed with TBS, and mounted using coverslips and Roti-Mount Aqua (Carl Roth (Cat. No. 2848)). Stainings were analyzed using the Leica fluorescence microscope DM6B-Z, equipped with the DFC3000G camera and the following filters: DAP (excitation filter [EX] 350/50, dichroic mirror [DC] 400, and a suppressor filter [SF] 460/50, L5 [EX 480/40, DC 505, SF 527/30], RHO [EX 546/10, DC 560, SF 585/40], Y5 [EX 620/60, DC 660, SF 700/76]; Leica). Leica application suite X (Leica) and ImageJ (National Institute of Health, Bethesda, MD, USA) were used for image analysis.

### 4.8. ScRNA-Seq

The mice of the second cohort were euthanized using CO_2_. The abdominal and thoracic cavities were opened, the inferior vena cava was cut, and blood was collected for serum analysis. A venipuncture cannula was placed into the left ventricle and the mice were shortly perfused with ice-cold Krebs solution. The colon and spinal cord were removed. Tissues from *n* = 3 mice, per treatment group, were pooled.

Colonic tissues were dissected, and fecal matter was removed by rinsing them with Krebs solution. The LMMP was dissected and transferred into a fresh tube containing Krebs solution. The tissues were washed three times by centrifugation at 400× *g* for 30 s and resuspension in Krebs solution. The tissues were minced using scissors and transferred into a fresh tube containing 9 mL Krebs solution. Then, 100 µL of a digestion solution containing 13 mg type II collagenase (Thermo Fisher (Cat. No. 17101015)) and 3 mg BSA was added. Tissues were digested in a shaking water bath at 37 °C for 1 h undergoing constant bubbling with 95% O_2_ and 5% CO_2_. Then, cells were centrifuged at 4 °C and 300× *g* for 7 min. The pellet was resuspended in 5 mL HBSS containing 0.05% trypsin (Thermo Fisher (Cat. No. 15090046)). Digestion was completed in a water bath for another 7 min. The trypsin reaction was stopped using 10 mL cold DMEM-F12 medium (Thermo Fisher (Cat. No. 11320074)) containing 10% FCS (Thermo Fisher (Cat. No. 10270166)) and 1% penicillin–streptomycin (Thermo Fisher (Cat. No. 15140122)) (rinse medium). The cells were centrifuged at 4 °C and 300× *g* for 8 min. The cells were then resuspended in rinse medium and passed through a 70-µm cell strainer. The strainer was rinsed with rinse medium and the cells were centrifuged at 4 °C and 300× *g* for 8 min. The pellet was resuspended in 1 mL rinse medium. The cell count and viability were determined. The cells were centrifuged one last time and resuspended in rinse medium. The cell concentration was adjusted to 1 × 10^6^ cells/mL, and the cells were sent to the Institute of Human Genetics at FAU Erlangen-Nürnberg for scRNA-seq.

The spinal cord was removed and transferred into RPMI-1640 without phenol red (Thermo Fisher (Cat. No. 11835)) with 10 mM HEPES (AppliChem (Cat. No. A1070); SC medium). The tissue was minced using scissors and centrifuged at 4 °C and 300× *g* for 10 min. The pellet was resuspended in 5 mL digestion solution containing 1 mg/mL type II collagenase (Worthington, Lakewood, CA, USA (Cat. No. LS004177)), 0.5 mg/mL type II dispase (Thermo Fisher (Cat. No. 17105041)), and 25 U/mL benzonase (Merck (Cat. No. 70746)). The tissue was digested in a shaking water bath at 37 °C for 30 min. In between, the cells were mechanically separated by pipetting up and down, 5× every 10 min. Digestion was stopped by adding 45 mL of cold PBS without Ca^2+^ and Mg^2+^ (Sigma-Aldrich (Cat. No. D8537)) with 1 mM EDTA. Samples were centrifuged at 4 °C and 300× *g* for 10 min. The pellet was resuspended in 1.8 mL PBS with 0.5% BSA (Carl Roth (Cat. No. CP84)) and 2 mM EDTA (=MACS buffer). Then, the samples were incubated with 200 µL myelin removal beads II (Miltenyi Biotec, Bergisch, Gladbach, Germany (Cat. No. 130-096-733)) for 15 min at 4 °C. Next, 20 mL of MACS buffer was added to the samples, which were then centrifuged at 4 °C and 300× *g* for 10 min. The pellets were then resuspended in 2 mL MACS buffer. The cells were separated using MACS LS columns (Miltenyi Biotec (Cat. No. 130-042-401)), which were prewetted with MACS buffer. The columns were rinsed with MACS buffer three times and the cell suspension was centrifuged. The cells were washed one time with 1 mL MACS buffer and centrifuged. The pellet was resuspended in 1 mL SC medium. The cell number and viability were determined. The cells were centrifuged, resuspended in SC medium, adjusted to a concentration of 1 × 10^6^ cells/mL, and sent to the Institute of Human Genetics at FAU Erlangen-Nürnberg for scRNA-seq.

For each sample, 8000 cells were subjected to 10× Chromium Single Cell 3′ Solution v3.1 library preparation according to the manufacturer’s instructions. Libraries were sequenced on the Illumina HiSeq 2500 sequencer (Illumina, San Diego, CA, USA) using the recommended read lengths for 10× Chromium v3.1 chemistry to a depth of 20,000 reads per cell. The reads were converted to the FASTQ format using *mkfastq* from Cell Ranger 4.0.0 (10× Genomics, Pleasanton, CA, USA). The reads were then aligned to the 10× Genomics mm10-2020-A mouse reference genome (mm10, Ensembl annotation release 98). The alignment was performed using the “count” command from Cell Ranger 4.0.0 (10× Genomics). Primary analysis, quality control filtering (gene count per cell, unique molecular identifier count per cell, and the percentage of mitochondrial ribosomal transcripts), clustering, uniform manifold approximation and projection (UMAP) projection, and visualization of gene expression were performed using the Seurat v3.2.0 package in R v4.0.2. Cells with less than 800 UMI or more than 20% (enteric nervous system) or 25% (spinal cord cells) mitochondrial genes were removed. Additionally, cells contaminated with immunoglobulin mRNA (>1%) were excluded from the analysis. The raw gene expression was scaled using the “SCTransform” approach. To compare the gene expression between conditions, SCTransform-scaled samples were integrated using the shared nearest neighbor approach as implemented in Seurat v3.2.0 [66]. Cluster marker genes were determined using edgeR v3.30.3. The edgeR model included cellular detection rate (=sequencing depth per cell) and sample identity as covariates to account for technical differences between samples. The same approach was used to find genes differentially expressed between cells under different conditions in one cell type. The edgeR model of differential expression test included the cellular detection rate as a covariate.

Heatmaps were created using GraphPad Prism version 9.3.1 (GraphPad Software Inc., San Diego, CA, USA) and show the top 10 differentially expressed genes in both directions for each treatment group, which means that the 10 genes with the highest and the 10 genes with the lowest normalized logFC values are shown in the heatmap. The modulus of 1.5 logFC (meaning >1.5 or <−1.5) was set as threshold for significant up- or downregulation, combined with an adjusted *p*-value of <0.05.

### 4.9. Statistical Analysis

For statistical analysis, GraphPad Prism version 9.3.1 was used. A two-way analysis of variance (ANOVA) was used to identify the differences between treatment groups when analyzing paired data (Figure 1C,D). For unpaired data (Figure 4 and Figure 7), one mean value per mouse calculated of duplicates was used for statistical analysis. Normality was determined by the Shapiro–Wilk test. One-way ANOVA was used for analyzing the differences between treatment groups. For qualitative ENS and spinal cord EM analysis (Figure 5B,C and Figure 8C,E,F), *n* = 5 images per mouse were analyzed. Values were used as replicates for a two-way ANOVA. For the quantitative EM analysis of the spinal cord, the g-ratio and subsequent interpretation of remyelinating axons (Figure 8B,D), mean values of 300 axons per mouse were used for a one-way ANOVA after the Shapiro–Wilk normality test. For all one-way ANOVA tests, Tukey’s post hoc test was used, when all groups passed the normality test. The Kruskal–Wallis post hoc test was used when the data did not follow normal distribution. *p*-values < 0.5 were statistically significant.

## Figures and Tables

**Figure 1 ijms-23-14209-f001:**
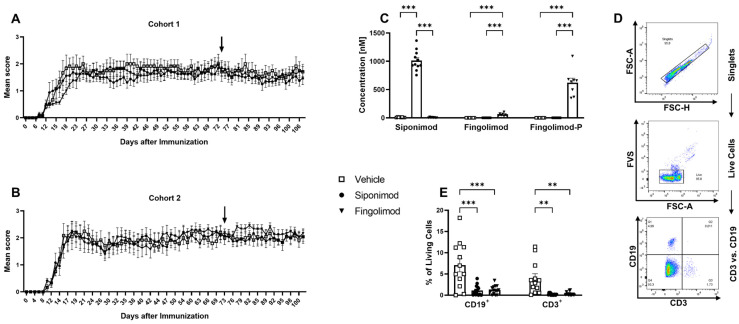
Development of experimental autoimmune encephalomyelitis in different treatment cohorts and validation of treatment success. Female C57BL/6J mice (9–12 weeks old) were immunized with MP4. (**A**,**B**) Disease course in two independent EAE cohorts. Arrows mark the beginning of siponimod or fingolimod treatment. (**C**) Thirty days after treatment initiation, serum samples were analyzed using mass spectrometry to determine siponimod, fingolimod, and fingolimod phosphate (fingolimod-P) concentrations. (**D**) Gating strategy for flow cytometry analysis of whole blood 30 d after treatment onset, which is shown in (**E**). Forward scatter height (FSC-H) vs. forward scatter area (FSC-A) was used to identify singlets. A fixable viability stain (FVS) was used to exclude dead cells. CD3 vs. CD19 comparison showed T (CD3^+^, CD19^−^), B (CD3^−^, CD19^+^), double negative, and double positive cells. Statistical analysis was performed using two-way ANOVA. ** *p* < 0.01; *** *p* < 0.001. EAE = experimental autoimmune encephalomyelitis, fingolimod-P = fingolimod phosphate, FCS-A = forward scatter area, FSC-H = forward scatter height, FVS = fixable viability stain, ANOVA = analysis of variance.

**Figure 2 ijms-23-14209-f002:**
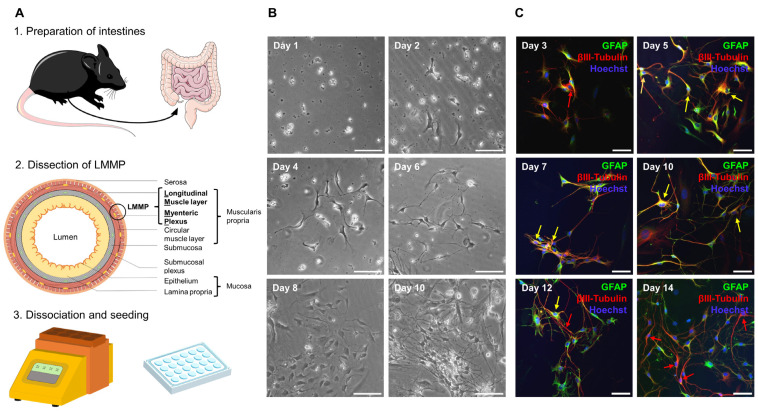
Establishment of a primary enteric nervous system culture. (**A**) Complete intestine from duodenum to rectum was removed, cleared of feces, and rinsed. Pieces that were 2–4 cm in length were threaded onto a glass rod and incised longitudinally. Then, the outer longitudinal muscle layer with attached myenteric plexus (LMMP) was dissected. LMMP was digested with collagenase and trypsin using the Miltenyi gentleMACS^TM^ Octo Dissociator with heaters. Cells were plated onto coated coverslips in a 24-well plate. The figure was partly generated using Servier Medical Art, provided by Servier, licensed under the Creative Commons Attribution 3.0 Unported license. (**B**) Cells began to adhere after 24 h. After 2 d, processes began to form and grow before cells started proliferating at approximately day 8. On day 10, a stable network had formed. Scale bars represent 100 µm. (**C**) Cells in culture expressed the marker proteins glial fibrillary acidic protein (GFAP) and βIII-tubulin. Immediately after culture preparation, few neuronal cells could be detected (red arrows). From day 5 onward, cells expressing both markers were visible (yellow arrows). From day 10 onward, coexpression decreased and more neuronal cells were detected. Scale bars represent 50 µm. ENS = enteric nervous system, GFAP = glial fibrillary acidic protein, LMMP = longitudinal muscle layer with attached myenteric plexus.

**Figure 3 ijms-23-14209-f003:**
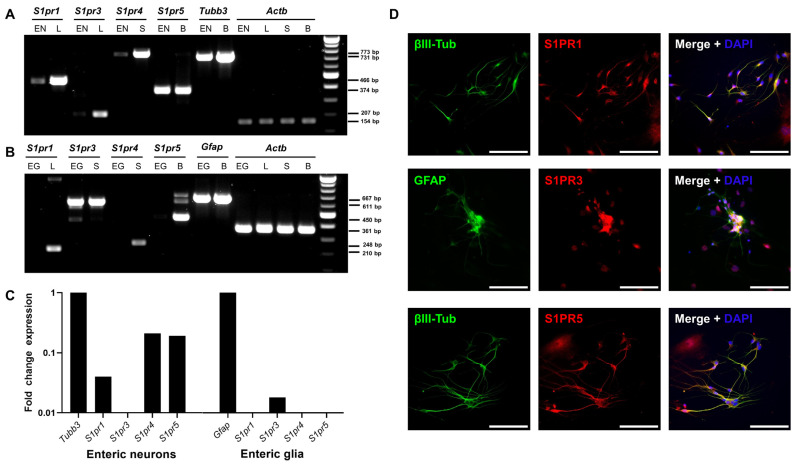
*S1pr* expression in the enteric nervous system. (**A**) RT-PCR was used to determine mRNA expression of S1P receptors in a murine enteric neuronal cell line (EN). As positive controls, the lung (L) was used for *S1pr1* and *S1pr3*, the spleen (S) for *S1pr4*, and the brain (B) for *S1pr5* and *Tubb3*, which served as a marker control for neuronal origin genes. (**B**) Expression of S1P receptors was determined in a rat enteric glial cell line (EG). *Gfap* was used as a control marker gene. (**C**) RT-PCR results were verified by qRT-PCR in EN and rat EG cell lines. Fold change expression was compared with the marker gene expression of *Tubb3* and *Gfap*. (**D**) Murine ENS primary cell cultures were stained for S1PR1, S1PR3 or S1PR5 in combination with the corresponding cell type markers GFAP or βIII-tubulin. Scale bars represent 100 µm. S1PR = sphingosine-1-phosphate receptor. EN = enteric neurons, L = lung, S = spleen, B = brain, EG = enteric glial cells, GFAP = glial fibrillary acidic protein, RT-PCR = reverse transcription polymerase chain reaction, qRT-PCR = quantitative real-time polymerase chain reaction.

**Figure 4 ijms-23-14209-f004:**
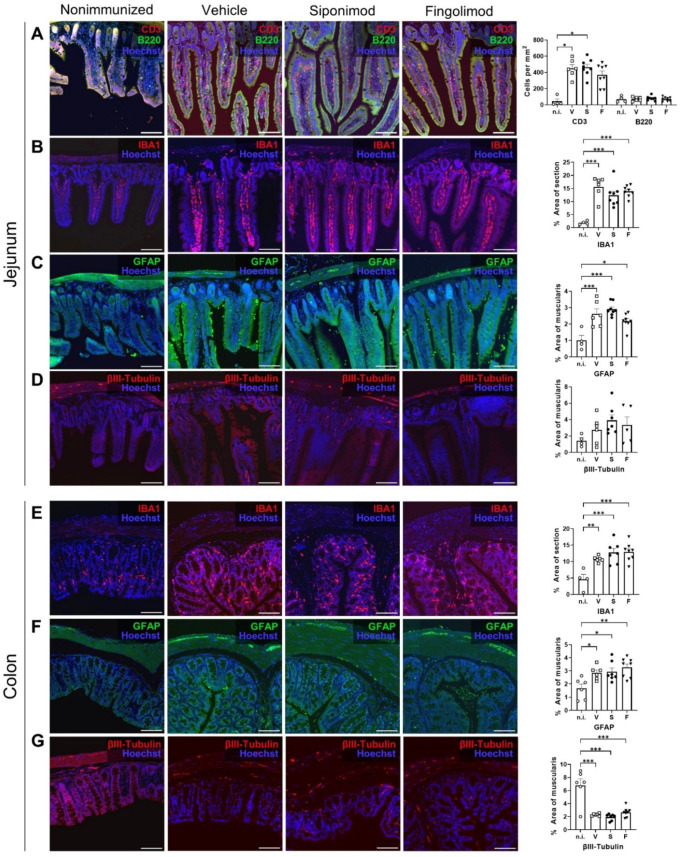
Immunohistochemical (IHC) staining of the jejunum and colon in siponimod- or fingolimod-treated experimental autoimmune encephalomyelitis (EAE) mice. IHC staining of (**A**–**D**) jejunum and (**E**–**G**) colon from nonimmunized (n.i.) versus chronic EAE mice treated with siponimod (S), fingolimod (F), or vehicle (V) and corresponding quantitative analysis. (**A**) Lamina propria infiltrating B and T cells were counted. (**B**,**E**) IBA1^+^ area was measured and compared with whole section area. (**C**,**F**) GFAP^+^ area was measured and compared with the muscularis area. (**D**,**G**) βIII-tubulin^+^ area was measured and compared with the muscularis area. Scale bars represent 100 µm. Statistical analysis was performed using one-way ANOVA with Tukey’s (IBA1, GFAP, and βIII-tubulin) or the Kruskal–Wallis (CD3) post hoc test. * *p* < 0.05; ** *p* < 0.01; *** *p* < 0.001. IBA1 = ionized calcium-binding adapter molecule 1, GFAP = glial fibrillary acidic protein, n.i. = nonimmunized, V = vehicle, S = siponimod, F = fingolimod, ANOVA = analysis of variance.

**Figure 5 ijms-23-14209-f005:**
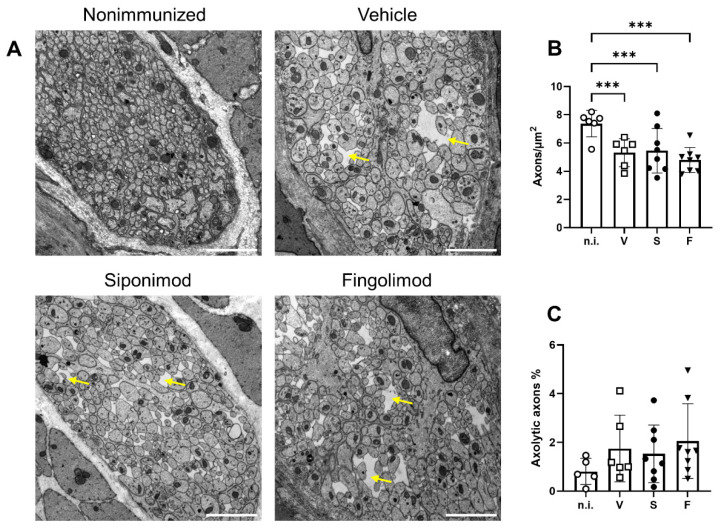
Analysis of the myenteric plexus in the colon of MP4-immunized mice using transmission electron microscopy. (**A**) Representative images of each treatment group in 10,000× magnification. Scale bars represent 2 µm. Yellow arrows indicate edematous gaps. (**B**) Quantification of the number of axons/µm^2^. (**C**) Quantification of the percentage of axolytic axons. Statistical analysis was performed using two-way ANOVA. *** *p* < 0.001. n.i. = nonimmunized, V = vehicle, S = siponimod, F = fingolimod, ANOVA = analysis of variance.

**Figure 6 ijms-23-14209-f006:**
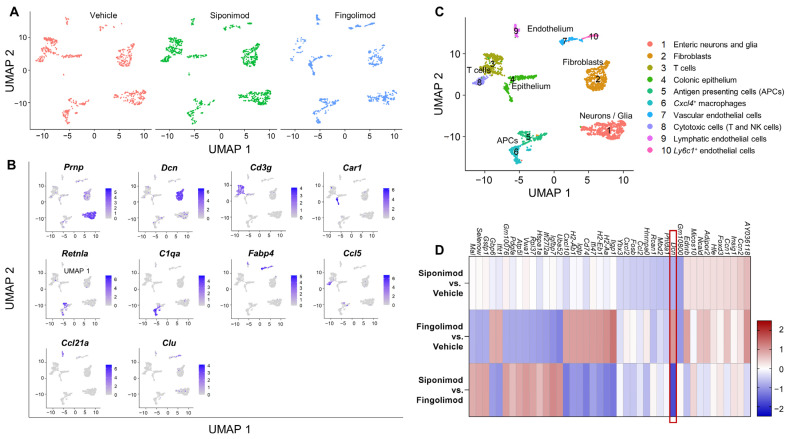
Single-cell RNA sequencing of the longitudinal muscle layer with attached myenteric plexus in chronic EAE mice. LMMP was digested mechanically and enzymatically, and mRNA from single cell suspensions was analyzed using scRNA-seq. (**A**) UMAP clustering with a resolution of 0.2 revealed 10 different clusters evenly detected in all treatment groups. (**B**) Top genes expressed by each cluster. (**C**) Clusters were classified according to marker genes. (**D**) Heatmap of differentially expressed mRNA between treatment groups of cluster 1. The heatmap shows the top 10 genes of each comparison with highest and lowest logFC values. The red box highlights the only gene with a logFC >|1.5|. LMMP = longitudinal muscle layer with attached myenteric plexus, scRNA-seq = single-cell RNA sequencing, logFC = log(fold change), UMAP = uniform manifold approximation and projection.

**Figure 7 ijms-23-14209-f007:**
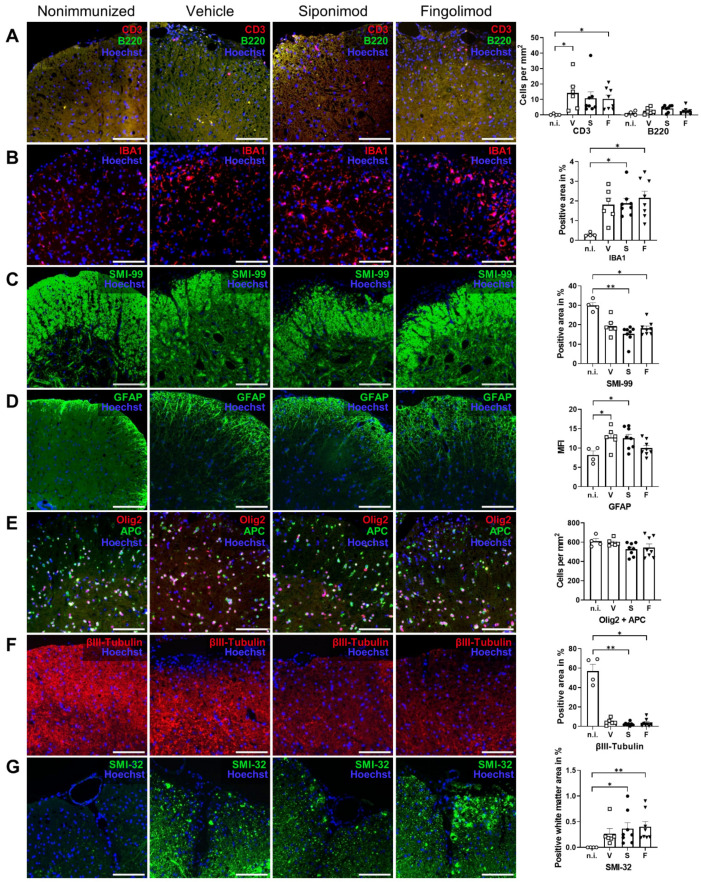
Immunohistochemical (IHC) staining of spinal cord sections of siponimod- or fingolimod-treated experimental autoimmune encephalomyelitis (EAE) mice. IHC staining of spinal cord sections from nonimmunized (n.i.) vs. chronic EAE mice treated with siponimod (S), fingolimod (F), or vehicle (V). Representative images of each treatment group and corresponding quantification. (**A**) Spinal cord infiltrating B and T cells were counted, (**B**) IBA1^+^ and (**C**) SMI-99^+^ area was measured and compared with the whole spinal cord section area, (**D**) mean fluorescence intensity (MFI) of GFAP staining was determined, (**E**) Olig2 + APC double-positive cells were counted, (**F**) βIII-tubulin^+^ area was measured and compared with the whole spinal cord section area, and (**G**) SMI-32^+^ area in the white matter was measured and compared with the white matter area. Scale bars represent 100 µm. Statistical analysis was performed using one-way ANOVA with either Tukey’s (GFAP, Olig2 + APC) or the Kruskal–Wallis (CD3 + B220, IBA1, SMI-99, βIII-tubulin, and SMI-32) post hoc test. * *p* < 0.05; ** *p* < 0.01. EAE = experimental autoimmune encephalomyelitis, GFAP = glial fibrillary acidic protein, MFI = mean fluorescence intensity, n.i. = nonimmunized, S = siponimod, F = fingolimod, V = vehicle, MBP = myelin basic protein, ANOVA = analysis of variance.

**Figure 8 ijms-23-14209-f008:**
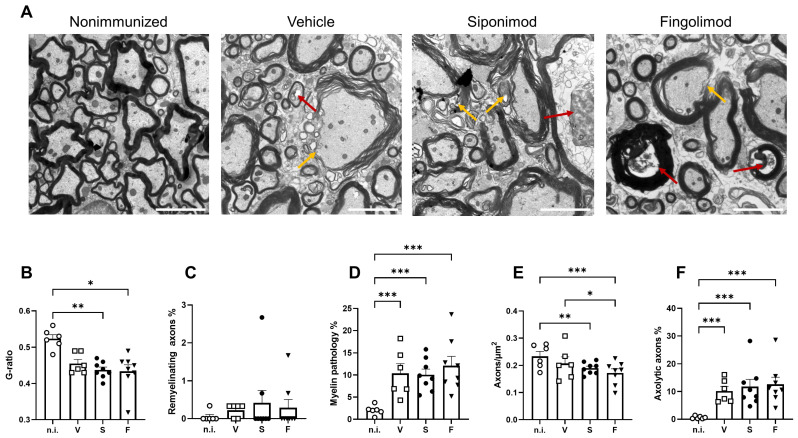
Transmission electron microscopy analysis of the spinal cord of siponimod- or fingolimod-treated experimental autoimmune encephalomyelitis (EAE) mice. (**A**) Representative images of all treatment groups. Red arrows indicate axolytic axons, and yellow arrows indicate myelin pathology. Scale bars represent 2 µm. Quantification of the (**B**) g-ratio, (**C**) percentage of axons with pathological myelin, (**D**) percentage of remyelinating axons, (**E**) axons/µm^2^, and (**F**) percentage of axons undergoing axolysis. Statistical analysis was performed using one-way ANOVA with either Tukey’s (myelin pathology, axons/µm^2^, and axolytic axons) or the Kruskal–Wallis (g-ratio and remyelinating axons) *post hoc* test. * *p* < 0.05; ** *p* < 0.01; *** *p* < 0.001. EAE = experimental autoimmune encephalomyelitis, n.i. = nonimmunized, S = siponimod, F = fingolimod, V = vehicle, ANOVA = analysis of variance.

**Figure 9 ijms-23-14209-f009:**
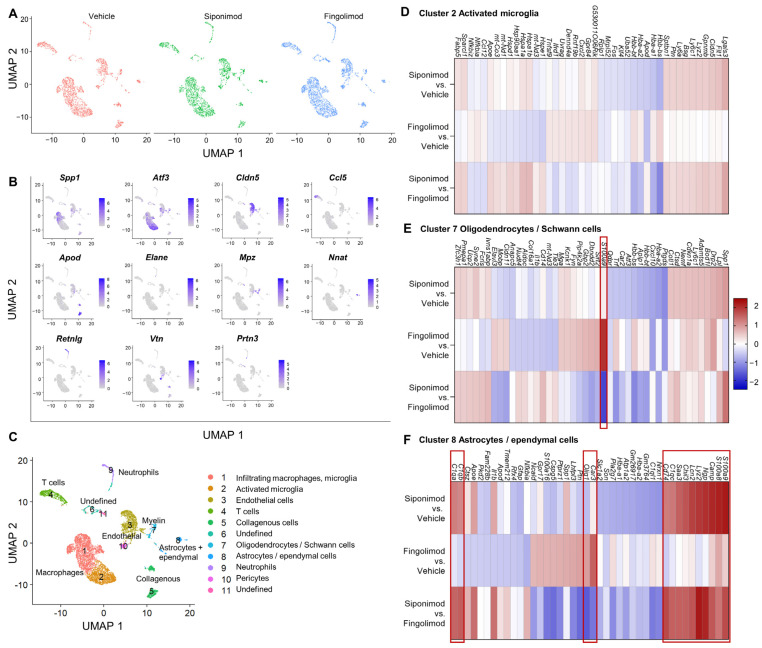
Single-cell RNA sequencing of spinal cord from chronic experimental autoimmune encephalomyelitis (EAE) mice. The spinal cord was digested mechanically and enzymatically, and mRNA from single cell suspensions was analyzed using scRNA-seq. (**A**) UMAP clustering with a resolution of 0.1 revealed 11 different clusters evenly detected in all three treatment groups. (**B**) Top genes expressed by each cluster. (**C**) Clusters were classified, according to marker genes. (**D**–**F**) Differentially expressed mRNA between treatment groups of clusters 2 (**D**), 7 (**E**) and 8 (**F**). Heatmaps show the 10 genes with the greatest difference in each comparison in both directions (up- and downregulation). Genes with a logFC >|1.5| are highlighted by red boxes. ScRNA-seq = single cell RNA sequencing, logFC = log(fold change); UMAP = uniform manifold approximation and projection.

**Table 1 ijms-23-14209-t001:** Clinical EAE parameters in various treatment groups.

	Cohort 1			Cohort 2			
Treatment	Vehicle	Siponimod	Fingolimod	Vehicle	Siponimod	Fingolimod	Total
Number of mice (*n*)	6	8	8	6	6	6	40
EAE onset (d.p.i.)	12.00 ± 1.03	11.75 ± 0.56	11.00 ± 0.65	12.00 ± 0.45	11.50 ± 1.02	12.33 ± 0.92	11.73 ± 0.30
Maximum EAE score	2.75 ± 0.28	2.63 ± 0.26	2.81 ± 0.27	2.79 ± 0.16	2.88 ± 0.20	2.88 ± 0.15	2.78 ± 0.09
Score before treatment	1.83 ± 0.21	1.56 ± 0.24	1.81 ± 0.27	2.21 ± 0.28	1.79 ± 0.16	2.29 ± 0.14	1.89 ± 0.10
Final score	1.71 ± 0.28	1.47 ± 0.24	1.72 ± 0.19	2.04 ± 0.25	2.04 ± 0.16	2.17 ± 0.17	1.83 ± 0.09
Score difference	−0.13 ± 0.15	−0.09 ± 0.25	−0.09 ± 0.18	−0.17 ± 0.23	+0.25 ± 0.13	−0.13 ± 0.15	−0.06 ± 0.08
Weight before treatment (g)	22.17 ± 0.92	22.36 ± 0.48	22.61 ± 0.43	22.85 ± 0.51	22.22 ± 0.32	22.22 ± 0.64	22.41 ± 0.22
Final weight (g)	22.77 ± 1.02	23.96 ± 0.67	22.96 ± 0.57	23.40 ± 0.48	23.72 ± 0.39	22.75 ± 0.64	23.28 ± 0.26
Weight difference (g)	+0.60 ± 0.40	+1.60 ± 0.52	+0.35 ± 0.25	+0.55 ± 0.31	+1.50 ± 0.31	+0.53 ± 0.26	+0.87 ± 0.163

Values are displayed as mean values ± standard error of the mean (SEM). EAE = experimental autoimmune encephalomyelitis, d.p.i. = days post immunization.

**Table 2 ijms-23-14209-t002:** LogFC values of genes that passed the > |1.5| threshold in cluster 8 of the spinal cord.

Gene Name	Protein Name	logFC_Siponimod/Vehicle_	logFC_Fingolimod/Vehicle_	logFC_Siponimod/Fingolimod_
*S100a9*	S100A9, Calprotectin	**2.25**	0.50	**1.75**
*S100a8*	S100A8, Calgranulin-A	**2.18**	0.85	1.33
*Camp*	Cethelicidin antimicrobial protein	**2.11**	0.48	*1.63 **
*Ngp*	Neutrophilic granule peptide	**2.04**	−0.10	*2.14 **
*Lyz2*	Lysozyme C2	**1.95**	−0.31	**2.26**
*Lcn2*	Lipocalin-2	**1.87**	0.20	**1.67**
*Chil3*	Chitinase-like protein 3	**1.67**	0.12	**1.5**
*Saa3*	Serum amyloid A-3 protein	**1.56**	0.01	**1.55**
*C1qc*	Complement C1q subunit C	1.32	−0.23	**1.55**
*Cd74*	CD74	1.29	−0.49	**1.78**
*Car3*	Carbonic anhydrase 3	0.19	**1.54**	−1.35
*Olig1*	Oligodendrocyte transcription factor 1	−0.66	0.95	**−1.61**
*C1qb*	Complement C1q subunit B	1.16	−0.42	**1.58**
*C1qa*	Complement C1q subunit A	1.25	−0.26	**1.52**

Values that passed the threshold of >|1.5| are written in bold. Values that passed the threshold, but did not have an adjusted *p*-value of <0.05 are written in italics and marked with *.

**Table 3 ijms-23-14209-t003:** Quality parameters used for mass spectrometry.

	Siponimod	Fingolimod	Fingolimod-P
Calibration Range	1–1000 ng/mL	0.1–1000 ng/mL	0.5–1000 ng/mL
LOQ	1 ng/mL	0.1 ng/mL	0.5 ng/mL
Accuracy (Bias)	<1.6%	<10.4%	<10.3%
Regression R^2^	0.99999	0.99925	0.99999
Precision RSD	±2.7%	±9.2%	±2.6%
Recovery	93.2%	87.8%	89.3%

**Table 4 ijms-23-14209-t004:** RT-PCR setup.

Initial Denaturation		95 °C	3 min
PCR (35 cycles)	Denaturation	95 °C	30 s
	Annealing	See Table 4	30 s
	Extension	72 °C	See Table 5
Final extension		72 °C	10 min

**Table 5 ijms-23-14209-t005:** Primers used for RT-PCR.

Species	Gene	Direction	Primer Sequence	Product Size	Annealing Temp.	Extension Time	Positive Control
*Mus* *musculus*	*Actb*	Forward	GGCTGTATTCCCCTCCATCG	154 bp	55 °C	12 s	-
		Reverse	TTGAGCGAGGCTGCTGTTTC				
	*Tubb3*	Forward	ATGAGGCCTCCTCTCACAAG	731 bp	56 °C	46 s	Brain
		Reverse	ATCGAACATCTGCTGCGTGA				
	*S1pr1*	Forward	TTGAGCGAGGCTGCTGTTTC	466 bp	57 °C	30 s	Lung
		Reverse	CGCCTGCTAATAGGTCCGAG				
	*S1pr3*	Forward	CTTCGGATTCTCTGGGGCAG	207 bp	56 °C	12 s	Lung
		Reverse	ATAGGCTCTCGTTCTGCAAGG				
	*S1pr4*	Forward	AGCCAATGGGCAGAAGTCTC	773 bp	57 °C	46 s	Spleen
		Reverse	ACAGTAGCCTGGGCATTGAC				
	*S1pr5*	Forward	CCGGTTACAGGAGACTTTTGC	374 bp	55 °C	22 s	Brain
		Reverse	ACAGTAGGATGTTGGTGGCG				
*Rattus norvegicus*	*Actb*	Forward	AGCCTTCCTTCCTGGGTATGG	361 bp	57 °C	22 s	-
		Reverse	GCAGCTCAGTAACAGTCCGC				
	*Gfap*	Forward	AACCGCATCACCATTCCTGT	667 bp	57 °C	40 s	Brain
		Reverse	TCTGCCCTACCCACTCCTAC				
	*S1pr1*	Forward	TTGAGCGAGGCTGCTGTTTC	210 bp	57 °C	15 s	Lung
		Reverse	AGCCTTAACCACTGGGATGC				
	*S1pr3*	Forward	GAACGAGAGCCTGTTTCCAAC	611 bp	56 °C	40 s	Spleen
		Reverse	TGCTTCTTGTTGGCGTCGTA				
	*S1pr4*	Forward	AACGGTTAGGCACAAGGAGG	248 bp	57 °C	15 s	Spleen
		Reverse	TTATGCTCAAGGTGCCCCAG				
	*S1pr5*	Forward	AGGCGCAAGGTTCGCATA	450 bp	56 °C	30 s	Brain
		Reverse	AGGAACATGGGTGCATGGAA				

**Table 6 ijms-23-14209-t006:** qRT-PCR setup.

UNG Incubation	Hold	50 °C	120 s
Polymerase Activation	Hold	95 °C	120 s
PCR (40 cycles)	Denaturation	95 °C	3 s
	Annealing/Extension	60 °C	30 s
Ramp Rate	4.4 °C/s		

**Table 7 ijms-23-14209-t007:** TaqMan Assays used for qRT-PCR.

Species	Gene	Catalog Number	Assay ID
*Mus musculus*	*Actb*	4453320	Mm01205647_g1
	*Tubb3*	4453320	Mm00727586_s1
	*S1pr1*	4453320	Mm00514644_m1
	*S1pr3*	4448892	Mm00515669_m1
	*S1pr4*	4448892	Mm00468695_s1
	*S1pr5*	4448892	Mm00474763_m1
*Rattus norvegicus*	*Actb*	4453320	Rn00667869_m1
	*Gfap*	4453320	Rn00566603_m1
	*S1pr1*	4448892	Rn00568869_m1
	*S1pr3*	4448892	Rn01757498_m1
	*S1pr4*	4448892	Rn01408095_s1
	*S1pr5*	4448892	Rn01486961_m1

## Data Availability

All data are available upon reasonable request.

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
