# Peer review of "Impact of Siponimod on Enteric and Central Nervous System Pathology in Late-Stage Experimental Autoimmune Encephalomyelitis"

_ijms, 2022, doi:10.3390/ijms232214209_

Round 1
Reviewer 1 Report
The authors used an EAE model to clarify the effects of siponimod and fingolimod. The experiments were very well conducted and robust and sophisticated methodologies were used. Although other studies provide data on siponimod and fingolimod being classically beneficial in the clinic, the authors did not observe a protective effect in their model. This is not a problem, as negative results, when well conducted, are important information for the scientific community.
I have only minor concerns:
- If the objective was to evaluate a chronic EAE model, why did the authors not use the SJL mice model?
- Why did they choose the oral route and not the systemic route through peritoneal inoculation?
- Considering this information: "Although early treatment was 505 successful, no positive effect of siponimod was observed when treatment started 30 d after 506 immunization [47].", Because the authors did not use a treatment in the initial moments, to be able to evaluate the effect during the development of the disease. In the C57BL/6 mouse EAE model, there is no remyelination. Thus, if there is an injury, little improvement will be observed. If the authors had carried out the treatment in the initial phase (>10 days) or even in the acute phase of the disease (> 15 days), they could have observed some effect, mainly on the immunological parameters. Therefore, I believe that the choice of treatment time was flawed for the animal model used (C57BL/6).
- About the conclusion: "results suggest that both siponimod and fingolimod are not ideal drug candidates for neuroprotective treatment in progressive MS.". The author need to revised this sentence. This statement can be maintained if restricted to siponimod considering a treatment in the chronic phase. But to say this about fingolimod, which is classically used and has several studies showing its effect, is wrong.
Author Response
We would like to thank this reviewer for the positive evaluation of our manuscript. In the following we provide a point-by-point response to the remaining comments.
1) If the objective was to evaluate a chronic EAE model, why did the authors not use the SJL mice model?
We thank the reviewer for this interesting question. In SJL mice, immunization with PLP139-151 leads to a relapsing-remitting disease course (McRae et al., J Neuroimmunol, 1992). Thus, this model is ideally suited to test, e.g., the ability of drugs to prevent relapses and to determine their effects on the peripheral immune response and how this immune response is associated with a relapse. However, one shortcoming of this model is that the time point of relapse and the duration of the relapse/remission stage varies in each mouse. Due to the unfluctuating disease course in the B6 model, a chronic disease time point can be more easily defined. Since it was our aim to study the effects of siponimod on late-stage disease to mirror SPMS we decided to rely on the B6 model, which is also well established in our animal facility.
2) Why did they choose the oral route and not the systemic route through peritoneal inoculation?
The study was sponsored by Novartis, which also provided us with siponimod- and fingolimod-loaded food pellets. Their clear recommendation that was based on their own in-house results was to use an oral route of administration. In addition, siponimod and fingolimod are approved as oral drugs for MS, so that we wanted to be as close to the human situation as possible.
3) Considering this information: "Although early treatment was 505 successful, no positive effect of siponimod was observed when treatment started 30 d after 506 immunization [47].", Because the authors did not use a treatment in the initial moments, to be able to evaluate the effect during the development of the disease. In the C57BL/6 mouse EAE model, there is no remyelination. Thus, if there is an injury, little improvement will be observed. If the authors had carried out the treatment in the initial phase (>10 days) or even in the acute phase of the disease (> 15 days), they could have observed some effect, mainly on the immunological parameters. Therefore, I believe that the choice of treatment time was flawed for the animal model used (C57BL/6).
It is known that remyelination predominantly occurs in the late disease stage in MP4-induced EAE in C57BL/6 mice. Comparing acute (peak disease), chronic (three months after EAE onset) and long-term (six months after EAE onset) disease, Prinz et al. showed significantly more remyelinating and simultaneously less demyelinating nerve fibers per mm2 in long-term disease compared to earlier time points (Prinz et al., PLoS One, 2015). Hence, remyelination indeed occurs in the B6 model, however only several weeks after EAE induction. This is why we have chosen the late time point for treatment start. In addition, it has already been demonstrated by multiple previous publications that siponimod and fingolimod are effective in earlier stages of EAE, mainly due to effects on the peripheral immune response (Gentile et al., J Neuroinflammation, 2016; Bail et al., J Neuroinflammation, 2017; Ward et al., JCI Insight, 2020; Brand et al., Neurology Neuroimmunol Neuroinflamm, 2022; Dietrich et al., Neurology Neuroimmunol Neuroinflamm, 2022). This is why we did not administer the drugs in the initial or acute stage of the disease. However, to accommodate this reviewer’s remark we have extended the discussion on page 16 of the revised manuscript.
4) About the conclusion: "results suggest that both siponimod and fingolimod are not ideal drug candidates for neuroprotective treatment in progressive MS.". The author need to revised this sentence. This statement can be maintained if restricted to siponimod considering a treatment in the chronic phase. But to say this about fingolimod, which is classically used and has several studies showing its effect, is wrong.
According to the reviewer’s suggestion we have modified the sentence on page 18 of the revised manuscript.
Reviewer 2 Report
In this manuscript, Weier et al. studied the effect of siponimod and fingolimod on enteric nervous system and central nervous system in a secondary progressive multiple sclerosis murin model. While the drugs are detectable in serum and cause the expected lymphopenia, authors couldn’t detect any effects in enteric or CNS histology on disease progression transcriptome of ENS and CNS cells. However, the study is well presented, well designed and polished. The authors studied the effect of the 2 drugs extensively and their analysis is sound. Even without positive results, the authors still reached interesting conclusion about the mode of action of siponimod and fingolimod. For these reasons, I suggest the manuscript should be accepted with minor modifications.
Introduction:
The introduction, while a bit long, allows the reader to understand all the points in the paper.
Results:
The authors should explain why the treatments started as late as 70 days post-immunization. Scose plateau was reached way earlier, couldn’t this consitute SPMS model? CNS damage shoudn’t be resversible this late in the disease. In the discussion, the authors explain that the late treatment is to remove the immune component of the disease in order to focus on the neurodegeneration aspect of the disease. However, we have no evidence in the text at which time point the immune component of the disease is remove and if an earlier treatment would have been possible.
Discussion:
The authors should develop more about the mechanisms underlying Dcn regulation. The others should develop on the proinflammatory genes associated with a worse MS outcome found in the scRNAseq and their lack of effect on the mouse disease progression.
Figures:
Figure 6C: In the UMAP, we should read ‘’astrocytes’’ instead of ‘’astrozytes’’. In the legend, ‘’?’’ could be replaced by ‘’undefined’’.
Author Response
We thank the reviewer for the positive evaluation of our manuscript. In the following, we will address the remaining minor concerns.
1) The authors should explain why the treatments started as late as 70 days post-immunization. Scose plateau was reached way earlier, couldn’t this consitute SPMS model? CNS damage shoudn’t be resversible this late in the disease. In the discussion, the authors explain that the late treatment is to remove the immune component of the disease in order to focus on the neurodegeneration aspect of the disease. However, we have no evidence in the text at which time point the immune component of the disease is remove and if an earlier treatment would have been possible.
We would like to thank the reviewer for this comment. Indeed, the score plateau was reached considerably before the onset of the treatment in our study. However, we have previously demonstrated that remyelination predominantly occurs in the late disease stage in MP4-induced EAE in C57BL/6 mice. Comparing acute (peak disease), chronic (three months after EAE onset) and long-term (six months after EAE onset) disease, Prinz et al. showed significantly more remyelinating and simultaneously less demyelinating nerve fibers per mm2 in the long-term disease compared to earlier time points (Prinz et al., PLoS One, 2015). This is why we have chosen the late time point for treatment start. We have extended the discussion accordingly on page 16 of the revised manuscript. In addition, we have modified our statement that there is no immune component in late-stage disease because this was indeed not quite correct.
2) The authors should develop more about the mechanisms underlying Dcn regulation. The others should develop on the proinflammatory genes associated with a worse MS outcome found in the scRNAseq and their lack of effect on the mouse disease progression.
Following the reviewer’s suggestion, we have included a discussion on Dcn regulation on pages 16 and 17 of the revised manuscript. In addition, we now elaborate further on the upregulation of proinflammatory genes in the siponimod group on page 18 of the revised manuscript.
3) Figure 6C: In the UMAP, we should read ‘’astrocytes’’ instead of ‘’astrozytes’’. In the legend, ‘’?’’ could be replaced by ‘’undefined’’.
Figure 6 shows single cell RNA sequencing results of the enteric nervous system. Hence, we believe that the reviewer rather refers to Figure 9. Figure 9C has been modified accordingly.